# The actin nucleation factors JMY and WHAMM enable a rapid Arp2/3 complex-mediated intrinsic pathway of apoptosis

**Virginia L. King**, **Nathan K. Leclair**, **Alyssa M. Coulter**, **Kenneth G. Campellone** *

Department of Molecular and Cell Biology, Institute for Systems Genomics, University of Connecticut, Storrs, Connecticut, United States of America

* kenneth.campellone@uconn.edu

## Abstract

The actin cytoskeleton is a well-known player in most vital cellular processes, but comparably little is understood about how the actin assembly machinery impacts programmed cell death pathways. In the current study, we explored roles for the human Wiskott-Aldrich Syndrome Protein (WASP) family of actin nucleation factors in DNA damage-induced apoptosis. Inactivation of each WASP-family gene revealed that two of them, *JMY* and *WHAMM*, are necessary for rapid apoptotic responses. JMY and WHAMM participate in a p53-dependent cell death pathway by enhancing mitochondrial permeabilization, initiator caspase cleavage, and executioner caspase activation. JMY-mediated apoptosis requires actin nucleation via the Arp2/3 complex, and actin filaments are assembled in cytoplasmic territories containing clusters of cytochrome *c* and active caspase-3. The loss of JMY additionally results in significant changes in gene expression, including upregulation of the WHAMM-interacting G-protein RhoD. Depletion or deletion of *RHOD* increases cell death, suggesting that RhoD normally contributes to cell survival. These results give rise to a model in which JMY and WHAMM promote intrinsic cell death responses that can be opposed by RhoD.

**Data Availability Statement:** All relevant data are within the manuscript and its Supporting Information files.

## Author summary

The actin cytoskeleton is a collection of protein polymers that assemble and disassemble within cells at specific times and locations. Cytoskeletal regulators called nucleation factors ensure that actin polymerizes when and where it is needed, and many of these factors are members of the Wiskott-Aldrich Syndrome Protein (WASP) family. Humans express 8 WASP-family proteins, but whether the different factors function in programmed cell death pathways is not well understood. In this study, we explored roles for each WASP-family member in apoptosis and found that a subfamily consisting of JMY and WHAMM is critical for a rapid pathway of cell death. JMY-mediated actin assembly in the cytoplasm is necessary for its pro-apoptotic function. Furthermore, the loss of JMY results in changes in gene expression, including a dramatic upregulation of the small G-protein RhoD, which appears to contribute to cell survival. Collectively, our results point to the

**Funding:** KGC was supported by National Institutes of Health grants R01-GM107441 and K02-AG050774 (www.nih.gov). The funders had no role in study design, data collection and analysis, decision to publish, or preparation of the manuscript.

**Competing interests:** The authors have declared that no competing interests exist.

importance of JMY and WHAMM in driving intrinsic cell death responses plus a distinct function for RhoD in maintaining cell viability.

## Introduction

Apoptosis is a programmed form of cell death crucial for many organismal processes, including development, tissue turnover, and tumor suppression [1,2]. It is driven by intrinsic mitochondria-mediated and extrinsic receptor-mediated death pathways that converge on a terminal execution program [3–6]. Cell rounding, shrinkage, membrane blebbing, and fragmentation into apoptotic bodies are common morphological features of apoptosis, and are controlled by a loss of actin-associated adhesions, rearrangements of actin filaments, and actin depolymerization [7,8]. While this reorganization and disassembly of the cytoskeleton during apoptosis is well described, the extent to which the actin assembly machinery actively contributes to the initiation or progression of apoptotic pathways is not understood.

Actin polymerization into filaments has been extensively characterized during cell morphogenesis, intracellular trafficking, cytokinesis, and motility [9,10]. To assemble branched actin networks during these processes, the heptameric Arp2/3 complex cooperates with nucleation-promoting factors from the Wiskott-Aldrich Syndrome Protein (WASP) family [11,12]. The mammalian WASP family is composed of WASP, N-WASP, WAVE1, WAVE2, WAVE3, WASH, WHAMM, JMY, and WHIMP [13,14]. Several of these factors are found in multi-protein complexes themselves, including all three WAVE isoforms and WASH, which constitute the WAVE and WASH complexes, respectively [15,16]. Each WASP-family member uses a conserved C-terminal domain that binds actin monomers and the Arp2/3 complex to stimulate polymerization, while their divergent N-terminal domains interact with a variety of small G-proteins and phospholipids which direct their different localizations and functions within cells [9,13]. N-WASP, WAVE1-3, and WHIMP drive plasma membrane protrusion; N-WASP and WASH control endocytosis and endosomal cargo sorting; WHAMM and JMY promote anterograde membrane transport and autophagy. Additionally, during most of these processes, the atypical nucleation-promoting factor Cortactin binds actin filaments and the Arp2/3 complex to modulate actin branchpoint stability [17].

Compared to their well-established roles in such vital cellular functions, the extent to which WASP-family proteins participate in cell death pathways is largely uncharacterized. WAVE1 is perhaps the most studied member in relation to apoptosis, where it can influence the localization or modification of Bcl-2-family proteins, which control mitochondrial outer membrane permeabilization and the release of apoptogenic proteins [6,18,19]. In hepatocytes, WAVE1 associates with the pro-apoptotic protein Bad [20], while in neuronal cells WAVE1 engages the anti-apoptotic protein Bcl-xL to promote mitochondrial recruitment of the pro-apoptotic permeabilization factor Bax [21]. In contrast, in leukemia cells, WAVE1 interacts with the anti-apoptotic protein Bcl-2 such that WAVE1 overexpression inhibits death signaling, whereas WAVE1 depletion increases apoptosis in response to chemotherapeutic agents [22,23]. Thus, WAVE1 appears to have cell type specific effects on apoptotic processes.

In addition to WAVE1, JMY has been reported to function in apoptosis. JMY was discovered as a cofactor that affects the function of p53 [24], a central tumor suppressor protein and transcription factor [25,26]. Under normal cellular growth conditions, JMY is regulated in the cytoplasm through interactions with the ubiquitin ligase Mdm2 [27] as well as autophagy-related proteins [28]. Upon genotoxic damage, JMY binds importins and accumulates in the nucleus [29,30], where it associates with the stress-response protein Strap, the histone

acetyltransferase p300, and p53 [24,31]. Co-overexpression of JMY and p300 in p53-proficient epithelial cells increases the transcription of *BAX* [24,29], while introduction of a JMY siRNA in p53-overexpressing cells decreases the amount of *BAX* induction [29]. These data indicate that JMY can promote cell death by enhancing the transcription of at least one pro-apoptotic gene. Consistent with this conclusion, an siRNA targeting JMY reduces the amount of sub-G1 (presumably apoptotic) cells after UV treatment [27]. Although, under other conditions, RNAi of JMY can increase the number of sub-G1 cells [32].

Collectively, these studies on WAVE1 and JMY reveal an interesting dynamic between actin assembly proteins and apoptosis-associated processes, and suggest that the relationships between nucleation factors and cell death pathways merit further investigation. In the current study, using a systematic gene knockout approach, we tested roles for all WASP-family members in DNA damage-induced apoptosis. We discovered that JMY and WHAMM are key regulators of a rapid p53-dependent and Arp2/3-mediated cell death pathway, but that their pro-apoptotic functions are opposed by the small G-protein RhoD.

## Results

### The WASP-family proteins JMY and WHAMM enable DNA damage-induced apoptosis

To evaluate the impact of WASP-family members on apoptosis, we employed a panel of knockout (KO) HAP1 or eHAP fibroblast-like cell lines lacking N-WASP, WAVE1, WAVE2, WAVE3, the WAVE Complex, the WASH Complex, WHAMM, JMY, or Cortactin (S1 Table and S1 Fig). The parental cells, derived from a chronic myeloid leukemia patient [33], are useful for studying apoptosis because they harbor a well-defined immortalizing BCR-ABL fusion [34], have easily manipulatable nearly-haploid (HAP1) or fully-haploid (eHAP) genomes [33,35], and are p53-proficient [36]. To examine intrinsic apoptotic responses, the parental and KO cells were treated with etoposide, a topoisomerase II inhibitor that induces dsDNA breaks. To monitor cell death, each sample was then imaged using fluorescent Annexin V (AnnV) to identify the apoptotic hallmark of phosphatidylserine externalization, propidium iodide (PI) to assess membrane permeability, and hoechst to visualize nuclear fragmentation (Figs 1A and S1).

After 6h of etoposide treatment, approximately 33% of parental HAP1 or eHAP cells were apoptotic, as determined by AnnV-positive staining, compared to only 3% of DMSO-treated controls (Fig 1B). The proportion of cells displaying AnnV staining was not significantly different across the majority of the etoposide- or DMSO-treated WASP-family knockout lines compared to the parental lines (Figs 1B and S1). In one of three etoposide experiments, a statistically significant increase in AnnV-positive cells was observed for the WAVE1[KO] line (Fig 1B), a phenotype reminiscent of previous studies in which WAVE1 depletion in leukemia cells increased apoptosis [22,23]. In contrast, across all three etoposide experiments, two cell lines—the JMY[KO] and the WHAMM[KO]—exhibited significantly lower percentages of AnnV-positive cells (Fig 1A and 1B). The JMY[KO] and WHAMM[KO] cells also showed substantially less frequent PI staining and nuclear fragmentation compared to parental cells (Figs 1A and S2). These results indicate that among a set of cell lines lacking WASP-family members, only those missing JMY or WHAMM have significantly impaired apoptotic responses to DNA damage.

JMY and WHAMM are approximately 35% identical and 50% similar to one another [37], and comprise a subgroup within the WASP family. Since both appear to be important for apoptosis, we sought to more thoroughly compare the effects of *JMY* or *WHAMM* inactivation on cell death. For this, we characterized three independent *JMY* mutant cell lines derived from HAP1 cells and two independent *WHAMM* mutant cell lines derived from eHAP cells (Figs

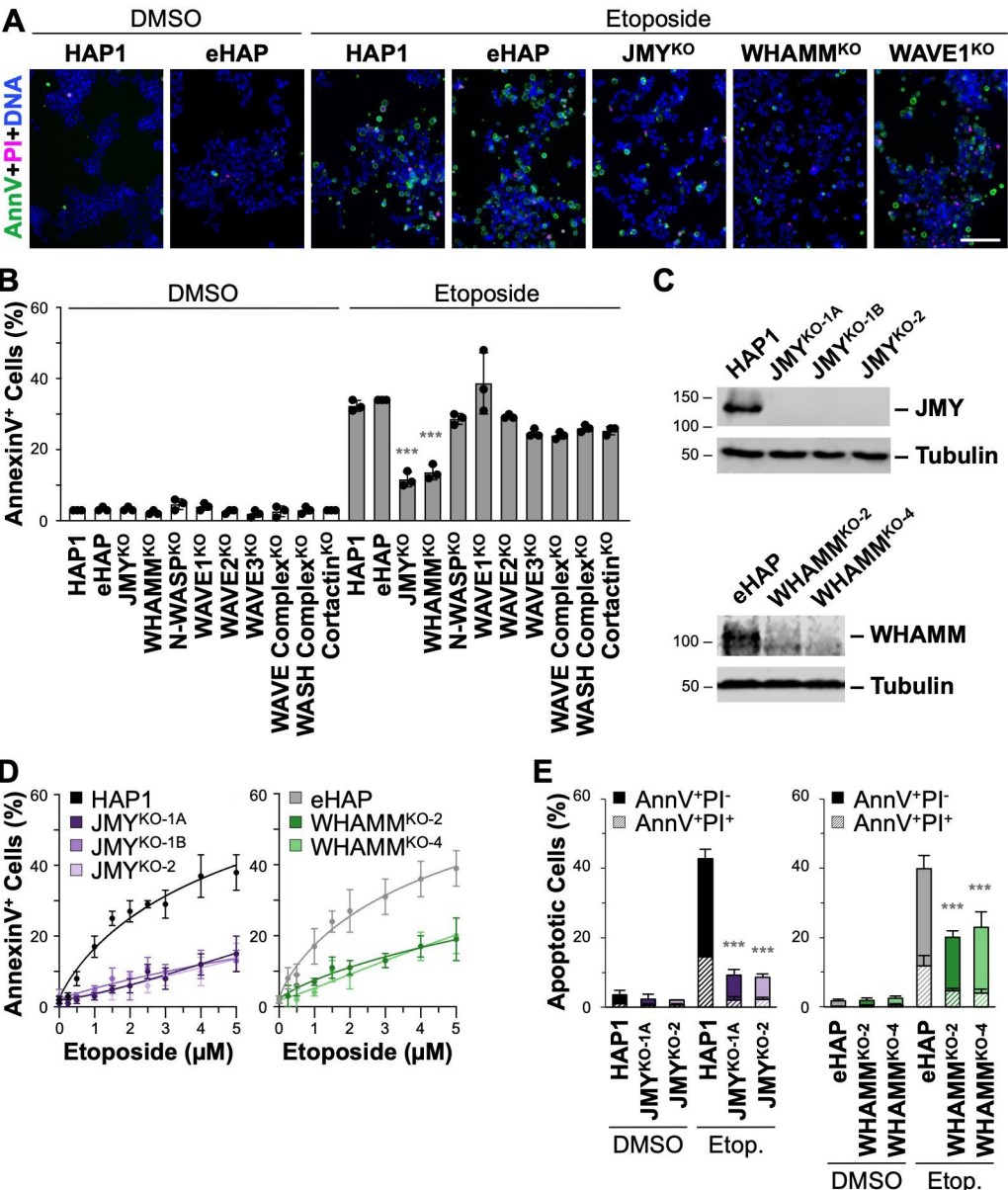

**Fig 1. Cells lacking the WASP-family members JMY or WHAMM undergo less apoptosis following DNA damage.**
**(A)** Parental (HAP1, eHAP) and WASP-family knockout (JMY[KO-1A], WHAMM[KO-2], N-WASP[KO], WAVE1[KO], WAVE2[KO], WAVE3[KO], WAVE Complex[KO], WASH Complex[KO], Cortactin[KO]) cells were treated with DMSO or 5μM etoposide for 6h and stained with Alexa488-AnnexinV (AnnV; green), Propidium Iodide (PI; magenta), and Hoechst (DNA; blue). Scale bar, 100μm. **(B)** The % of AnnV-positive cells was calculated in ImageJ by dividing the number of cells that displayed AnnV staining by the total number of cells identified by nuclear Hoechst staining. Each point represents the mean from 3 fields-of-view in 3 separate experiments, and the bar represents the mean ±SD of those experiments (n = 391–1,303 cells per point; n = 1,772–3,619 cells per bar). The uppermost WAVE1[KO] point was significantly different from HAP1 cells in one experiment. **(C)** HAP1, JMY[KO], eHAP, and WHAMM[KO] cell lysates were immunoblotted with antibodies to JMY, WHAMM, and tubulin. **(D)** HAP1, JMY[KO], eHAP, and WHAMM[KO] cells were treated with a range of etoposide concentrations for 6h and then stained with Alexa488-AnnV and Hoechst. The % of AnnV-positive cells was calculated in ImageJ. Each point represents the mean ±SD from 3–6 fields-of-view pooled from 1–2 experiments (n = 1,699–4,996 cells per experiment). Nonlinear regressions were performed with a baseline set to 0.02 and a maximum response set to 0.85. EC50 values were significantly different for parental vs KO samples (Mean EC50s: Parental = 6.2μM; JMY[KO] = 26.4μM, p<0.001; WHAMM[KO] = 20.9μM, p<0.001). **(E)** Cells were treated with DMSO or 5μM etoposide for 6h and stained with Alexa488-AnnV, PI, and Hoechst. The % of AnnV-positive cells is displayed as the fraction of AnnV-positive/PI-negative (AnnV+PI-) or AnnV/PI double-positive (AnnV+PI+) cells. Each bar represents the mean ±SD from 3 experiments (n = 2,338–9,846 cells per bar). All significance stars are for comparisons to the etoposide-treated HAP1 or eHAP cells. ***p<0.001 (ANOVA, Tukey post-hoc tests).

1C and S3). Etoposide treatment caused similarly high levels of DNA damage across all of the parental and knockout samples, as evidenced by increased phosphorylated histone H2AX (γH2AX) protein levels and greater numbers of γH2AX foci at DNA breaks within nuclei (S4 Fig). Both the parental cell lines, as well as the JMY$^{KO}$ and the WHAMM$^{KO}$ lines, showed etoposide dose-dependent increases in AnnV-positive apoptotic cells, but the mutants displayed significantly lower percentages at virtually every concentration (Fig 1D). At the highest concentrations (4–5μM), nearly 40% of parental cells, but only 8–12% of JMY$^{KO}$ cells and 15–20% of WHAMM$^{KO}$ cells, were AnnV-positive (Fig 1D).

To assess early versus late apoptosis, the percentage of AnnV-positive cells without versus with PI staining was quantified (S2 Fig). Among the ~40% of parental cells exhibiting AnnV staining, about 25% were AnnV-positive and PI-negative, indicating early apoptosis, while 15% were AnnV and PI double-positive, signifying late apoptosis (Fig 1E). In comparison, all of the JMY$^{KO}$ and WHAMM$^{KO}$ samples contained significantly fewer cells in both early and late apoptosis, with the JMY$^{KO}$ cells exhibiting 8% early and 2% late, and the WHAMM$^{KO}$ cells displaying 15% early and 5% late (Fig 1E). These results indicate that JMY and WHAMM each play important roles in apoptosis, but the more extreme phenotypes observed when *JMY* is mutated suggest that JMY is more prominent in enabling cell death.

## Inactivation of *JMY* or *WHAMM* impairs multiple steps of intrinsic apoptotic signaling

Since the loss of JMY or WHAMM decreased the final apoptotic readouts of phosphatidylserine externalization and membrane permeability, we next wanted to determine if the inactivation of *JMY* or *WHAMM* affected earlier aspects of apoptotic signaling. Intrinsic pathways of apoptosis are characterized by the export of apoptogenic proteins from mitochondria into the cytosol upstream of the initiation of a caspase cleavage cascade. So we examined whether the loss of JMY or WHAMM affected the release of cytochrome *c* (cyto *c*), a protein that is maintained in the mitochondrial intermembrane space under normal conditions, but is exported to trigger the cytosolic assembly of macromolecular structures called apoptosomes, which serve as platforms for the activation of initiator caspases during intrinsic apoptosis [38,39]. In parental HAP1 and eHAP cells, as expected, mitochondrial-independent cyto *c* localization became apparent following etoposide treatment (Fig 2A and 2B). Categorization of cyto *c* localization as discrete mitochondrial versus diffuse cytosolic was confirmed by plotting the fluorescence intensity profiles of mitochondria and cyto *c* (Fig 2C and 2D). It was also notable that diffuse cytosolic cyto *c* staining was sometimes accompanied by non-mitochondrial clusters of cyto *c* puncta (described in a later section). Consistent with the previously-observed amounts of AnnV-positive cells, >30% of parental cells exhibited cytosolic cyto *c* localization patterns following 6h of etoposide treatment (Fig 2E). In contrast, the JMY$^{KO}$ and WHAMM$^{KO}$ cell lines had significant delays in cyto *c* release, and by 6h only 10% of JMY$^{KO}$ and around 15% of WHAMM$^{KO}$ cells showed cytosolic cyto *c* staining (Fig 2E). These findings suggest that JMY and WHAMM are necessary for an efficient intrinsic apoptosis pathway, and that part of their contributions to apoptotic responses may occur prior to and/or during the accumulation of cyto *c* in the cytosol.

After mitochondrial cyto *c* release, apoptotic signaling is typified by the multimerization and proteolytic processing of initiator caspases in the apoptosome that cleave and activate executioner caspases, which in turn target multiple proteins in the cytosol and nucleus [40,41]. We next sought to measure the effects of JMY and WHAMM on the caspase cascade by examining both the expression and cleavage of representative initiator and executioner caspase proteins via immunoblotting. Control DMSO-treated parental, JMY$^{KO}$, and WHAMM$^{KO}$ cells

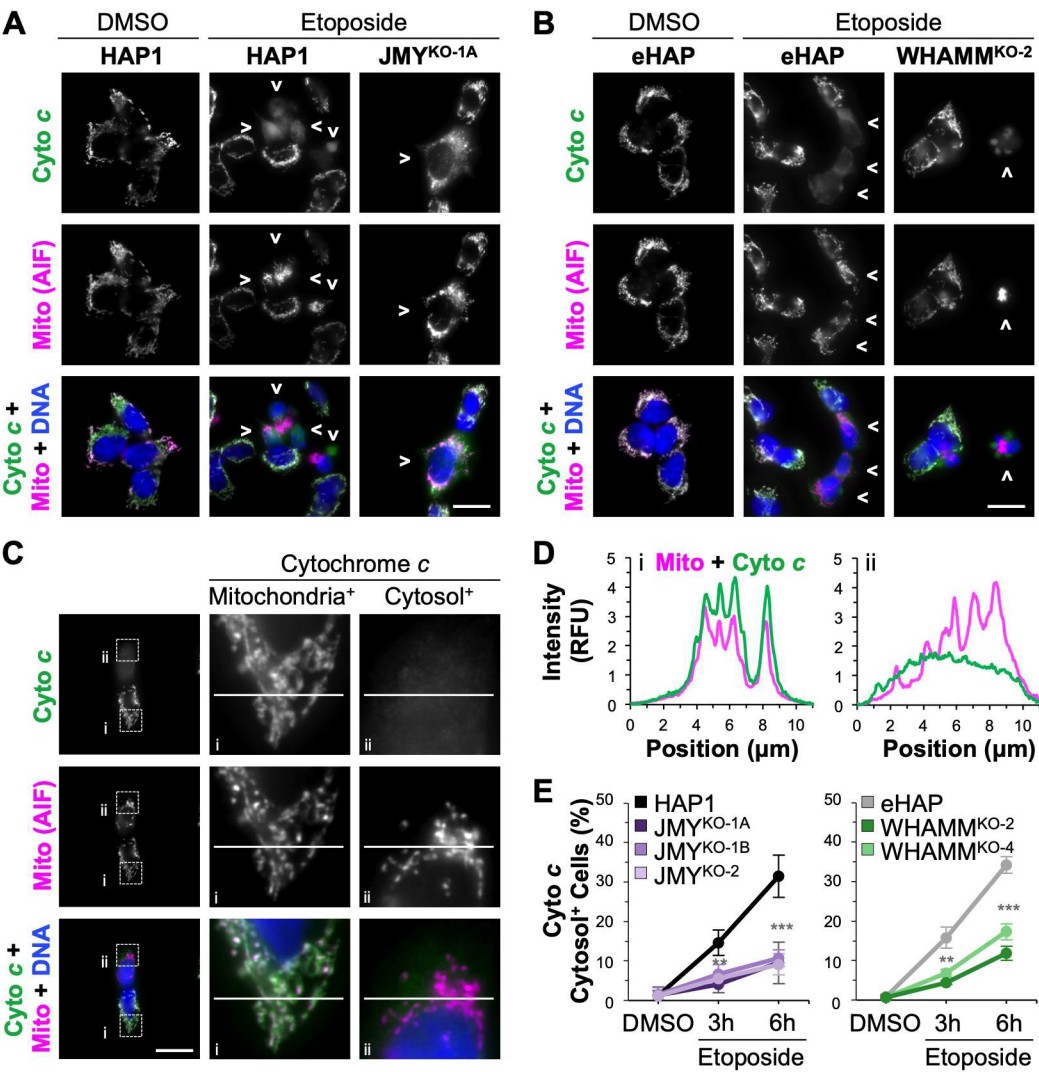

**Fig 2. Cytochrome *c* release is delayed when JMY or WHAMM is deleted. (A-B)** HAP1, JMY[KO], eHAP, and WHAMM[KO] cells were treated with DMSO for 6h or 5μM etoposide for 3h or 6h before being fixed and stained with a cytochrome *c* antibody (Cyto *c*; green), AIF antibody (Mito; magenta), and DAPI (DNA; blue). Images are from the 6h timepoint. Arrowheads highlight examples of cells with diffuse cytosolic cyto *c* staining. Scale bars, 25μm. **(C)** A representative image of HAP1 cells with magnifications depicts cyto *c* in the mitochondria (i) or diffuse in the cytosol (ii). **(D)** Lines were drawn through the magnified images using ImageJ to measure the pixel intensity profiles for cyto *c* and mitochondria, highlighting the differences in scoring mitochondrial versus cytosolic cyto *c* localization. **(E)** The % of cells with cytosolic cyto *c* staining was calculated in ImageJ by counting cells that exhibited cytosolic rather than mitochondrial cyto *c* staining and dividing by the total number of DAPI-stained nuclei. Each point represents the mean % ±SD from 3–4 experiments (n = 533–856 cells per point). Significance stars refer to comparisons of parental to KO samples at the depicted timepoints. **p<0.01; ***p<0.001 (ANOVA, Tukey post-hoc tests).

expressed indistinguishable levels of inactive initiator procaspase-9 and inactive executioner procaspase-3 (Fig 3A and 3B), suggesting that the mutant cell lines did not have any inherent deficits in the steady-state abundance of these core components of the apoptotic machinery. However, upon etoposide treatment for 3h and 6h, the parental, JMY[KO], and WHAMM[KO] cell lines displayed very different kinetics in the conversion of inactive procaspases to cleaved caspases. Whereas HAP1 and eHAP cells began to show processing of procaspases to cleaved caspases by 3h of etoposide treatment, the cleaved caspases were generally not detectable in the

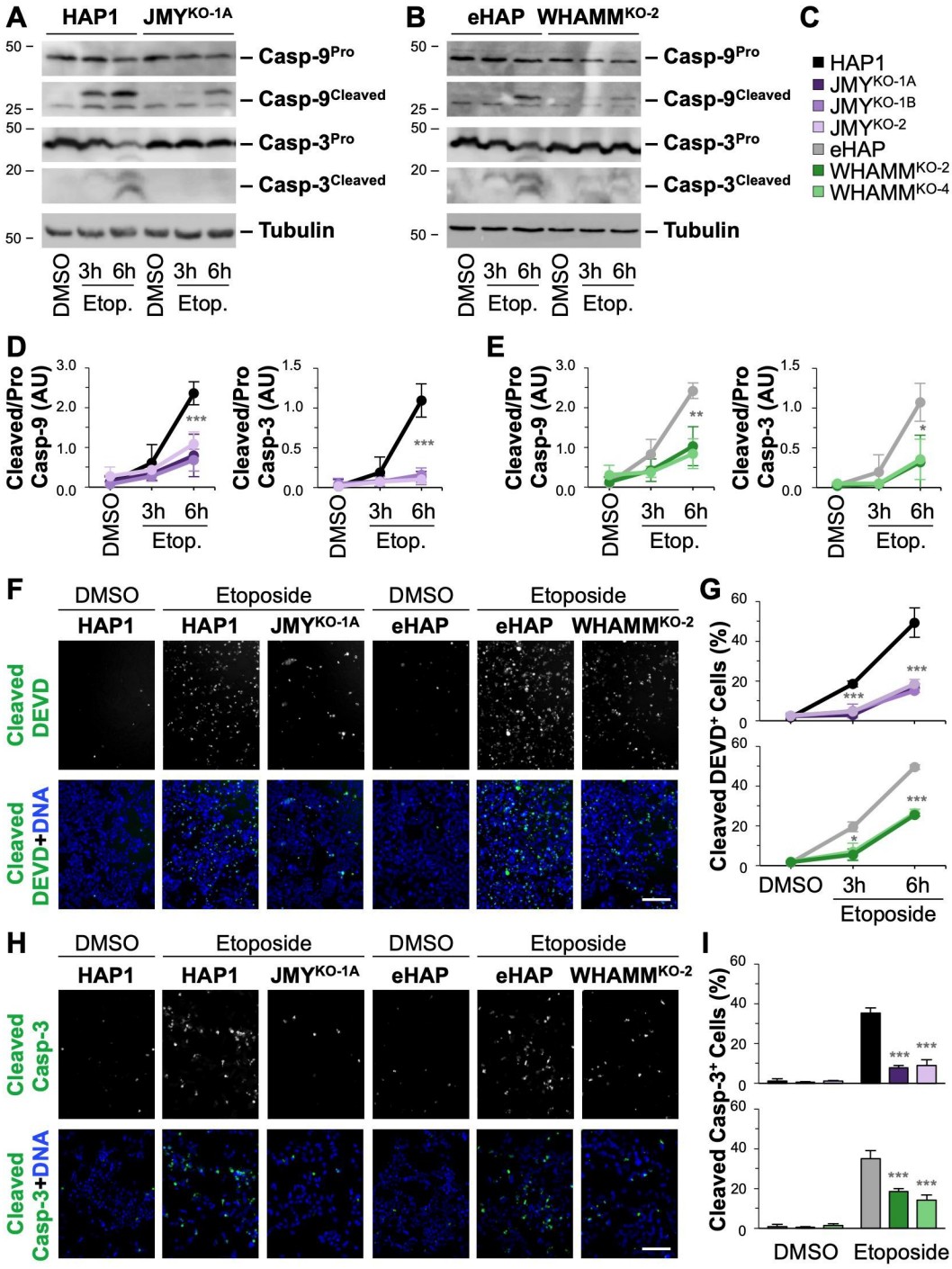

**Fig 3. Initiator and executioner caspase cleavage is inefficient in the absence of JMY or WHAMM. (A-B)** HAP1, JMY$^{KO}$, eHAP, and WHAMM$^{KO}$ cells were treated with DMSO for 6h or 5μM etoposide for 3h or 6h, and extracts were immunoblotted with antibodies to caspase-9 (Casp-9$^{Pro}$ and Casp-9$^{Cleaved}$), caspase-3 (Casp-3$^{Pro}$ and Casp-3$^{Cleaved}$), and tubulin. **(C)** Key for color-coded samples in panels (D-I). **(D-E)** For quantification, the caspase cleavage ratio was calculated by dividing the cleaved band intensity by the procaspase band intensity. Each point represents the mean ratio ±SD from 3 experiments. AU = Arbitrary Units. **(F)** Cells were treated with DMSO or etoposide, and stained with caspase-3/7 green detection reagent to label cleaved DEVD (green) and Hoechst to stain DNA (blue). Images are from the 6h timepoint. Scale bars, 100μm. **(G)** The % of cleaved DEVD-positive cells was calculated in ImageJ by counting cells that exhibited green nuclear fluorescence and dividing by the total number of Hoechst-stained nuclei. Each point represents the mean ±SD from 3 experiments (n = 2,741–6,646 cells per point). **(H)** Cells treated as above were fixed and stained with an antibody that

recognizes caspase-3 cleaved at Asp175 (Cleaved Casp-3; green) and DAPI (DNA; blue). Images are from the 6h timepoint. **(I)** The % of cleaved caspase-3-positive cells was calculated in ImageJ by counting cells that contained bright clusters of green fluorescence and dividing by the total number of DAPI-stained nuclei. Each point represents the mean ±SD from 3 experiments (n = 1,136–7,844 cells per bar). Significance stars refer to comparisons of parental to KO samples. $^*p<0.05$; $^{**}p<0.01$; $^{***}p<0.001$ (ANOVA, Tukey post-hoc tests).

JMY$^{KO}$ and WHAMM$^{KO}$ cells until 6h (Fig 3A and 3B). Even at the later timepoint, caspase-9 cleavage was 2-3-fold lower in both JMY$^{KO}$ and WHAMM$^{KO}$ cells compared to the parental cell lines (Fig 3C–3E). For caspase-3, cleavage was 10-fold less efficient in JMY$^{KO}$ cells and 3-fold less efficient in WHAMM$^{KO}$ cells relative to their parental lines (Fig 3C–3E). Together, these results show that following acute genotoxic damage, JMY and WHAMM are each required for the rapid activation of caspase cleavage cascades.

During the terminal execution stage of apoptosis, caspase-3 and caspase-7 cleave the motif Asp-Glu-Val-Asp (DEVD) [41], so we used a DEVD substrate peptide conjugated to a fluorescent reporter to quantify the amount of cells with active caspase-3/7 following treatment with etoposide. Caspase-3/7 activation was readily detectable in parental cells at 3h, but did not become clearly apparent in the knockout cells until the 6h timepoint, when 50% of parental cells versus only 15–25% of JMY$^{KO}$ or WHAMM$^{KO}$ cells were positive for cleaved DEVD (Fig 3F and 3G). To further validate these results, we stained cells with antibodies that specifically recognize the active cleaved form of caspase-3. Immunofluorescence microscopy revealed that nearly 40% of parental cells contained bright clusters of active caspase-3 staining, whereas <10% of the JMY$^{KO}$ and <20% of the WHAMM$^{KO}$ cells stained positive for active caspase-3 (Fig 3H and 3I). Therefore, the apoptotic deficiencies arising from *JMY* or *WHAMM* mutations include significantly delayed and less potent activation of executioner caspases, with JMY knockout cells exhibiting more extreme defects, particularly in caspase-3 activation.

## Cells lacking JMY and/or WHAMM are defective at apoptosis in several cellular contexts

Because the loss of either JMY or WHAMM results in multiple apoptotic deficits, we next sought to determine the extent to which apoptosis would proceed in cells lacking both proteins. So we derived two independent WHAMM/JMY double knockout (DKO) cell lines from WHAMM$^{KO-2}$ cells (S3 Fig). Similar to the other knockout cells, both WHAMM/JMY$^{DKO}$ cell lines had low levels of DNA damage when exposed to DMSO, but more numerous DNA breaks when incubated with etoposide (S4 Fig). After etoposide exposure and staining with fluorescent AnnV, significantly fewer WHAMM/JMY$^{DKO}$ cells were apoptotic compared to control cells, as only 10% of WHAMM/JMY$^{DKO}$ cells were AnnV-positive (Fig 4A and 4B). This frequency of apoptosis was noticeably lower than that of the single WHAMM knockouts and more closely resembled that of the single JMY knockouts (Fig 1E).

To directly compare the efficiency of apoptosis in the parental cells, single *JMY* or *WHAMM* mutants, and *WHAMM/JMY* double mutants, we stained each of these cell populations for active cleaved caspase-3. While nearly 40% of parental HAP1 or eHAP cells contained active caspase-3 after etoposide treatment, 20% of WHAMM$^{KO}$ cells and <10% of JMY$^{KO}$ or WHAMM/JMY$^{DKO}$ cells showed active caspase-3 staining (Fig 4C and 4D). Thus, inactivation of *JMY* in a *WHAMM*-deficient background does not result in any synthetic or synergistic death phenotypes, but simply reduces apoptosis levels down to that of a single *JMY* mutant. This frequency, typically 8–10% of the population, seems to be the lowest achievable amount of apoptotic death in HAP1 derivatives under the genotoxic conditions used in our experiments.

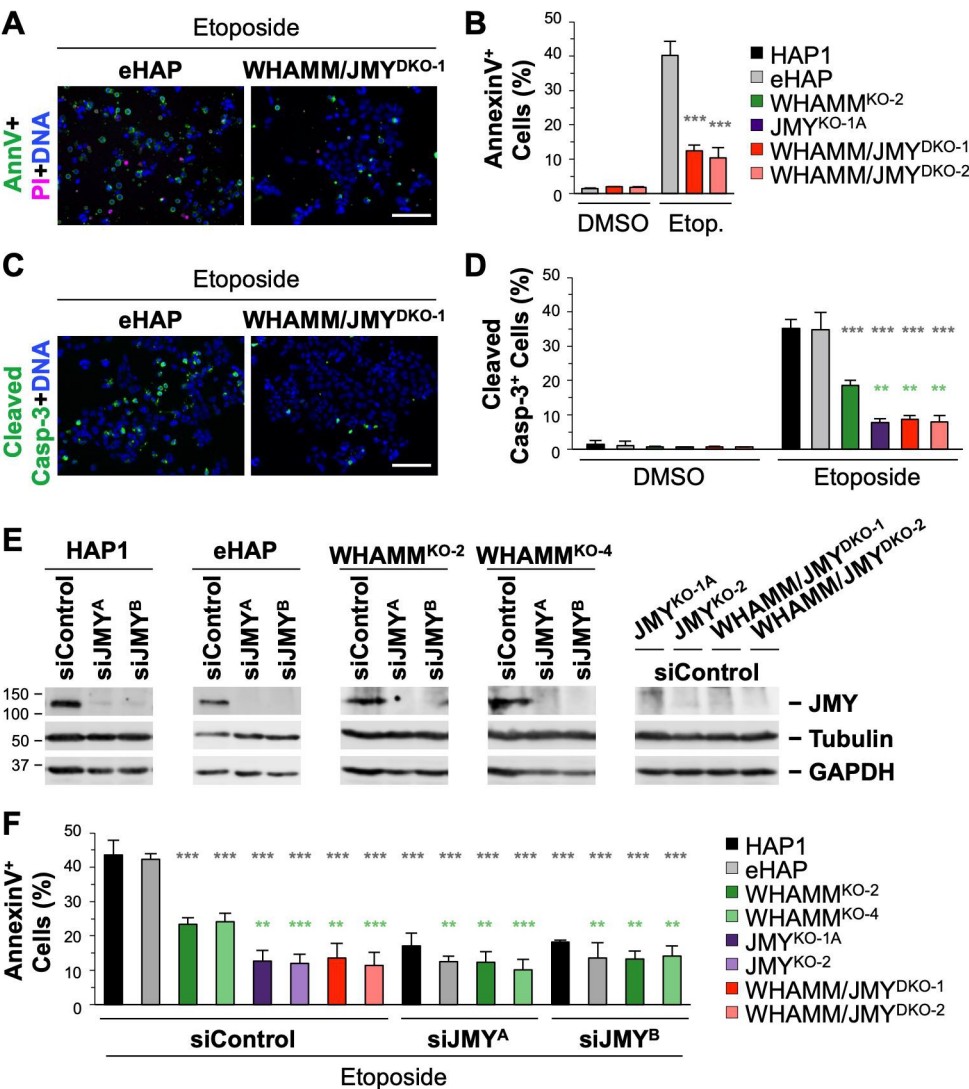

**Fig 4. Deletion or depletion of JMY reduces apoptosis in WHAMM-deficient cells. (A)** eHAP and WHAMM/JMY$^{DKO}$ cells were treated with DMSO or 5μM etoposide for 6h and stained with Alexa488-AnnV, PI, and Hoechst. Scale bars, 100μm. **(B)** The % of AnnV-positive cells was calculated, and each bar represents the mean ±SD from 3 experiments (n = 3,035–7,080 cells per bar). **(C)** eHAP and WHAMM/JMY$^{DKO}$ cells were treated with DMSO or etoposide before being fixed and stained with an antibody that recognizes caspase-3 cleaved at Asp175 (Cleaved Casp-3; green) and DAPI (DNA, blue). **(D)** The % of cleaved caspase-3-positive cells was calculated, and each bar represents the mean ±SD from 3 experiments (n = 2,962–7,844 cells per bar). Gray significance stars refer to comparisons to the parental samples and green significance stars refer to comparisons to the WHAMM$^{KO}$ samples. **(E)** HAP1, eHAP, and WHAMM$^{KO}$ cells were treated with control siRNAs or independent siRNAs for the JMY gene, while JMY$^{KO}$ and WHAMM/JMY$^{DKO}$ cell lines were treated with control siRNAs before immunoblotting with antibodies to JMY, tubulin, and GAPDH. **(F)** Etoposide-treated cells were stained with Alexa488-AnnV, PI, and Hoechst. The % of AnnV-positive cells was calculated and each bar represents the mean ±SD from 3 experiments (n = 2,485–4,596 cells per bar). Gray significance stars refer to comparisons to the parental siControl samples and green significance stars refer to comparisons to the WHAMM$^{KO}$ siControl samples. **p<0.01; ***p<0.001 (ANOVA, Tukey post-hoc tests).

To next determine the extent to which transient depletion, rather than permanent deletion, of JMY, WHAMM, or both, affects apoptosis, we treated HAP1 and eHAP cells with two independent siRNAs targeting either JMY (Fig 4E) or WHAMM (S5 Fig). Consistent with the stable genetic knockout results, significantly fewer JMY- or WHAMM-depleted cells were AnnV-positive compared to parental cells after etoposide exposure (Figs 4F and S5). Moreover,

RNAi-mediated depletion of JMY in WHAMM$^{KO}$ cells resulted in significantly fewer AnnV-positive cells compared to control siRNA treatments (Fig 4E and 4F), a phenotype resembling that of the WHAMM/JMY$^{DKO}$ lines. Overall, the more severe apoptotic deficiencies that arise when JMY is permanently deleted or transiently depleted suggest that JMY is the nucleation factor with the most influential role in programmed cell death.

For further gauging the effects of JMY on apoptosis in different human cell lines, we targeted JMY using siRNAs in U2OS osteosarcoma cells and HeLa adenocarcinoma cells (S5 Fig). As in HAP1 and eHAP cells (Figs 4E, 4F, 5A, and 5B), RNAi-mediated depletion of JMY resulted in significantly fewer AnnV-positive cells compared to control siRNA treatments (S5 Fig). Additionally, the level of JMY expressed in HAP1, U2OS, and HeLa cells showed a positive relationship with the proportion of AnnV-positive cells (Figs 5C and S5). These results demonstrate that JMY is required for efficient apoptosis across multiple cell types.

## JMY functions in p53-dependent apoptosis after a p21-associated proliferation arrest

To begin to understand the underlying mechanisms that give rise to the apoptotic pathway defects in JMY- and WHAMM-deficient cells, we sought to determine the importance of p53 in the death of parental HAP1 and eHAP cells. First, to verify that apoptosis was dependent on p53, we treated these parental cell lines with three independent siRNAs targeting *TP53* and measured the percentage of cells with AnnV staining. Compared to samples that were treated with a control siRNA, samples in which p53 levels were diminished contained significantly fewer AnnV-positive cells (Figs 5D, 5E, and S6). Moreover, greater degrees of p53 knockdown resulted in larger reductions in apoptosis, as the amount of p53 protein was positively correlated with the percentage of AnnV-positive cells (Figs 5F and S6). Extrapolation of the p53 trendline down to zero protein expression suggested that approximately 12% of HAP1 cells completely lacking p53 would undergo apoptosis (Figs 5F and S6), nearly matching the percentage observed for JMY-depleted samples (Fig 5C). These results demonstrate that the majority of etoposide-induced apoptosis in HAP1 cells requires both p53 and JMY.

For evaluating the impact of p53 levels on apoptosis in the absence of JMY or WHAMM, we then targeted *TP53* in WHAMM$^{KO}$ and JMY$^{KO}$ cells. Depletion of p53 in WHAMM$^{KO}$ cells resulted in significantly fewer AnnV-positive cells (S6 Fig), consistent with the idea that *WHAMM* inactivation causes only a partial inhibition of p53-dependent apoptosis which can be rendered complete by the subsequent removal of p53. Conversely, depletion of p53 in the *JMY* mutant background revealed that the already low apoptosis levels following etoposide treatment could not be reduced below 10% (Fig 5D–5F). Together, the JMY and p53 depletion studies support the conclusion that JMY and p53 function in the same intrinsic apoptotic death pathway.

The observation that HAP1 cells undergo apoptosis in a primarily p53-dependent manner, combined with the finding that JMY can associate with p53 and its acetyltransferase cofactor p300 [24], led us to next investigate the specific influence of JMY on p53 functions. Since apoptosis is typically accompanied by p53 protein stabilization, post-translational modification, and re-localization into the nucleus [25,26], we examined these properties. At steady-state, HAP1 and JMY$^{KO}$ cells expressed similar amounts of p53 protein (S7 Fig), indicating that the loss of JMY did not cause an overall decrease in p53 abundance. Treatment of HAP1 and JMY$^{KO}$ cells with etoposide resulted in a clear increase in p53 protein levels, its phosphorylation on serine-15 and serine-46, its acetylation on lysine-382, and its accumulation in the nucleus in the majority of cells (S7 Fig), suggesting that JMY functions downstream of such pro-apoptotic changes in p53 stability and modification.

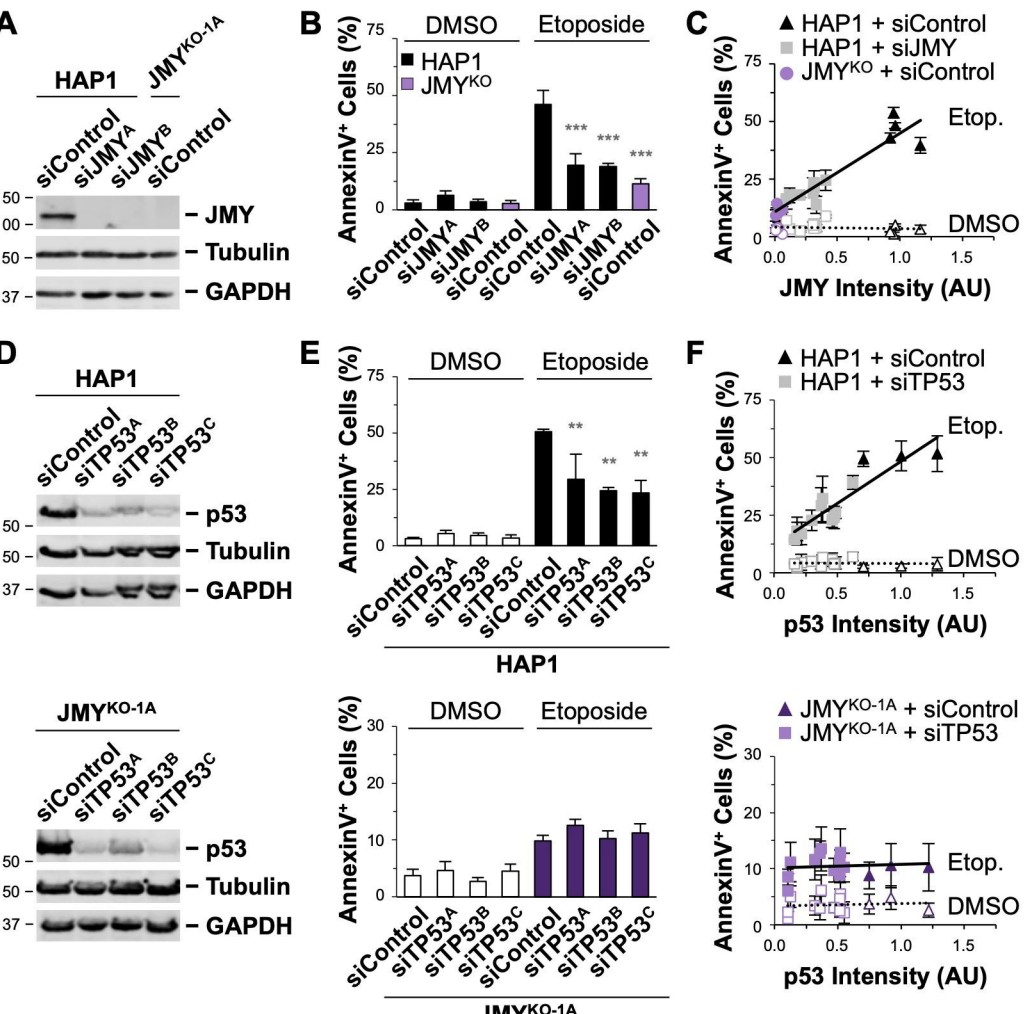

**Fig 5. Cells transiently depleted of JMY or p53 have similar reductions in apoptosis.** (A) HAP1 cells were treated with control siRNAs or independent siRNAs for the JMY gene, while a JMY[KO] cell line was treated with control siRNAs before immunoblotting with antibodies to JMY, tubulin, and GAPDH. (B) Cells were treated with DMSO or 5μM etoposide for 6h and stained with Alexa488-AnnV, PI, and Hoechst. The % of AnnV-positive cells was calculated, and each bar represents the mean ±SD from 4 experiments (n = 4,508–7,626 cells per bar). (C) JMY band intensities on immunoblots were normalized to tubulin intensities and plotted against the % of AnnV-positive cells. Each point represents the mean ±SD from 3 fields-of-view in a given experiment (n = 564–2,587 cells per point). The slope in the linear trendline regression equation for etoposide-treated samples (Y = 34.56X + 10.52) was significantly non-zero (p<0.001, R$^2$ = 0.86). (D) HAP1 and JMY[KO-1A] cells were treated with control siRNAs or independent siRNAs for the TP53 gene before immunoblotting with antibodies to p53, tubulin, and GAPDH. (E) Cells were treated and stained as in panel (B). The % of AnnV-positive cells was calculated, and each bar represents the mean ±SD from 3 experiments (n = 2,194–5,237 cells per bar). (F) p53 band intensities on immunoblots were normalized to tubulin intensities and plotted against the % of AnnV-positive cells. Each point represents the mean ±SD from 3 fields-of-view in a given experiment (n = 273–2,217 cells per point). The slope in the linear trendline regression equation for HAP1 etoposide-treated samples (Y = 33.47X + 12.48) was significantly non-zero (p<0.001, R$^2$ = 0.74). Significance stars refer to comparisons to the HAP1 siControl samples. **p<0.01; ***p<0.001 (ANOVA, Tukey post-hoc tests).

In response to genotoxic damage, nuclear p53 alters transcription and can trigger cell cycle arrest, DNA repair, apoptosis, senescence, and other stress responses [42,43]. Because a proliferation arrest is usually an early response to DNA damage, we compared the growth and death rates of HAP1 and JMY[KO] cultures after exposure to etoposide. For these experiments, we treated each cell line with DMSO or etoposide for 6h, removed the solvent or drug and

replaced them with fresh media, and then quantified the numbers of total cells (live and dead) at regular intervals up to a 48h endpoint. While HAP1 and JMY$^{KO}$ cells multiplied at equivalent rates after treatment with DMSO, both cell types stopped proliferating after treatment with etoposide (S8 Fig). In contrast, when apoptotic cell quantities were measured using AnnV staining, the proportion of cells undergoing apoptosis was significantly higher in the HAP1 samples at every timepoint (S8 Fig). For parental samples, >35% were AnnV-positive by 6h, 50% were apoptotic by approximately 10h, and the frequency of apoptosis had reached about 90% of cells by 48h (S8 Fig). JMY$^{KO}$ samples never hit the 50% apoptosis mark, as only 35% were AnnV-positive at 48h (S8 Fig). Thus, without JMY, cells remain stuck in an arrested condition and fail to shift into a proper death signaling state.

One of the key p53 targets that triggers cell cycle arrest is *CDKN1A*, which encodes the cyclin-dependent kinase inhibitor, p21 [44]. To determine if *CDKN1A* expression is affected by *JMY* inactivation in the absence or presence of etoposide, we used RT-PCR for comparing *CDKN1A* transcript levels. Consistent with the proliferation assays described above, *CDKN1A* was expressed at low levels in DMSO-treated HAP1 and JMY$^{KO}$ cells, and significantly upregulated in both cell lines following a 6h exposure to etoposide (S8 Fig). Similarly, small amounts of p21 protein were found in HAP1 and JMY$^{KO}$ cells at steady-state, while treatment with etoposide resulted in a 3-fold increase in the abundance of nuclear p21 in both sets of cells (S8 Fig). Collectively, these experiments suggest that JMY specifically promotes a cell suicide program and is not required for several other aspects of nuclear p53 modification or function, including the transcriptional responses that lead to a proliferation arrest.

## Expression of the small G-protein RhoD is turned on in JMY-knockout cells

While inactivation of *JMY* did not prevent arrest or *CDKN1A* upregulation, it could still impact other aspects of transcriptional programming. Indeed, given that JMY can enhance the expression of the pro-apoptotic p53 target *BAX* [24,29], one plausible explanation for the apoptotic defects in JMY$^{KO}$ cells could be that they possess different gene expression patterns than normal cells such that they are less 'equipped' to die. To characterize the transcriptomic changes that took place upon knocking out *JMY*, we performed differential gene expression analyses using RNA-sequencing (RNA-seq) on HAP1 cells and on one of the JMY knockout cell lines (JMY$^{KO-1A}$). When comparing the JMY$^{KO-1A}$ cells to parental HAP1 cells, <0.36% of protein-coding genes displayed expression differences of at least 2-fold and with a significance q-value of <0.05 (Figs 6A and S9). Expression of the genes encoding p53, caspases, Bax and other Bcl-2-family members, additional key modulators of apoptosis, or JMY-interacting proteins such as p300, Strap, and Mdm2 were not significantly different (S10 Fig). Moreover, genes for WASP-family members and other actin nucleation factors were not substantially changed (S10 Fig). Therefore, mutating *JMY* does not appear to adversely affect the basal expression of canonical apoptosis regulators or factors that are structurally or functionally related to JMY.

Since important differences in gene expression might not become evident until cells experience a pro-apoptotic stimulus, we next compared the transcriptomic changes that occurred in HAP1 and JMY$^{KO}$ cells after etoposide exposure. When comparing etoposide-treated to control samples, <0.37% of protein-coding genes in either cell line displayed expression differences of at least 2-fold and with a significance q-value of <0.05 (Figs 6B, 6C, and S9). Of the 67 total genes that met such criteria, 48 were shared between the HAP1 and JMY$^{KO}$ cells (S9 Fig). The mutually-upregulated factors included the *CDKN1A* cell cycle inhibitor described above, NFκB signaling components, and AP-1 transcription factors (S9 Fig). Substantial changes in

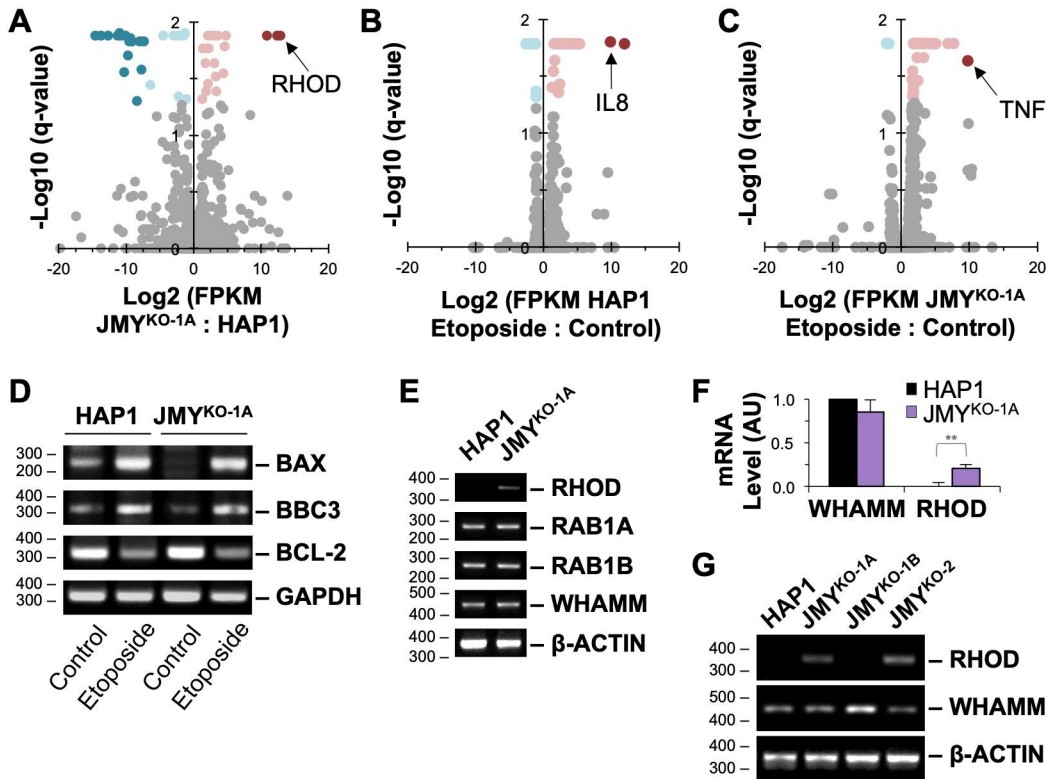

**Fig 6. Expression of *RHOD*, but not canonical apoptosis regulators, is turned on in JMY-knockout cells. (A)** RNA collected from HAP1 and JMY^{KO-1A} cells was subjected to mRNA sequencing analysis. A volcano plot represents FPKM (Fragments Per Kilobase of transcript per Million mapped reads) values plotted against -log10(q-values) of all individual genes for JMY^{KO-1A} vs HAP1 cells. Genes with expression differences of at least 2-fold and with a significance q-value of <0.05 are shown in dark red (turned on), pink (up-regulated), light blue (down-regulated), or dark blue (turned off), while genes that do not meet these criteria are depicted in gray (unchanged). Each point represents the mean value from 3 independent RNA samples per genotype. **(B-C)** RNA collected from HAP1 and JMY^{KO-1A} cells treated with 5μM etoposide for 6h was subjected to mRNA sequencing. Volcano plots represent FPKM values plotted against -log10(q-values) of all individual genes for etoposide vs control media-treated HAP1 (B) or JMY^{KO} (C) cells. Genes with expression differences of at least 2-fold and with a significance q-value of <0.05 are shown as in A. Each point represents the mean value from two independent RNA samples per genotype, except for *IL8*, which was only detected in one experiment. **(D-E)** RNA from HAP1 and JMY^{KO} cells was subjected to RT-PCR using primers to *BAX*, *BBC3*, *BCL-2*, *GAPDH*, *RHOD*, *RAB1A*, *RAB1B*, *WHAMM*, and *β-ACTIN*, and visualized on agarose gels. **(F)** Gel band intensities were quantified in ImageJ, and values for *RHOD* and *WHAMM* were normalized to *β-ACTIN*. AU = Arbitrary Units. Each bar represents the mean ±SD from 3 experiments. **(G)** RNA from HAP1, JMY^{KO-1A}, JMY^{KO-1B}, and JMY^{KO-2} cells were subjected to RT-PCR and visualized on agarose gels. **p<0.01 (t-test).

expression of genes that encode p53, caspases, Bcl-2 family members, or other prototypical apoptotic modulators were again not detected. Among the 19 genes that showed statistical differences between the parental and knockout cells, 16 displayed similar trends in either up- or down-regulation. The only sizable discrepancies between the two cell lines were in the cytokine genes *IL8* and *TNF*, as *IL8* was strongly upregulated in one of the HAP1 samples and *TNF* was greatly upregulated in the JMY^{KO} samples (Figs 6B, 6C, and S9). These results imply that HAP1 and JMY^{KO} cells are sufficiently equipped with enough potentially-apoptotic factors to be able to execute an intrinsic death program without a major increase in the abundance of pro-apoptotic transcripts.

A closer inspection of known p53 targets in the RNA-seq datasets indicated that *BAX* transcript levels were unchanged after etoposide treatment in both HAP1 and JMY^{KO} cells (S10 Fig). Under the same genotoxic conditions, the *BBC3* transcript, which encodes the p53

upregulated modulator of apoptosis PUMA, was increased 2.5-fold in HAP1 cells and 5-fold in the JMY^{KO}, although neither of these values reached statistical significance (S10 Fig). To more specifically test whether changes in the amounts of particular transcripts could be detected by a targeted approach, we performed RT-PCRs for *BAX* and *BBC3*, as well as the anti-apoptotic factor *BCL-2*. Compared to untreated steady-state conditions, etoposide exposure resulted in slightly increased *BAX* and *BBC3* transcript levels and decreased *BCL-2* levels in both HAP1 and JMY^{KO-1A} cells (Fig 6D). Therefore, mutating *JMY* does not appear to drastically affect the expression patterns of canonical p53 target genes, further suggesting that transcriptional reprogramming is not the principal driver of death signaling under the conditions used in our experiments.

In light of these observations, we sought to better understand how unanticipated factors might be contributing to cell survival or death in the presence and absence of JMY. When examining the 51 genes whose expression differed in the JMY^{KO} line at steady state, it was notable that the most upregulated gene was *RHOD* (Figs 6A and S9), which encodes a small G-protein that was previously shown to interact with WHAMM [45,46]. Transcript levels for other small G-proteins were virtually the same between parental and KO cells (S10 Fig), implying that the change for *RHOD* might be functionally meaningful. Moreover, RT-PCR experiments confirmed that *RHOD* transcript levels were almost undetectable in HAP1 cells and significantly turned on in the JMY^{KO} cells (Fig 6E and 6F). The expression of *WHAMM* and *RAB1A/RAB1B*, the latter of which encode small G-proteins known to bind WHAMM directly [47], were unchanged between the parental and JMY^{KO} cells (Figs 6E and S10). RT-PCR further showed that one of the other two independently derived *JMY* knockout cell lines also had increased *RHOD* transcript levels (Fig 6G), suggesting that *RHOD* upregulation may be a frequent, but not universal, response to the inactivation of *JMY*. These experiments reveal that although the loss of JMY may not cause any obvious compensatory changes in the transcription of actin nucleation factors, it can result in increased expression of a G-protein known to affect cytoskeletal dynamics.

## RhoD promotes cell survival in the absence or presence of JMY

RhoD is a Rho-family GTPase that regulates many cellular functions, including actin assembly during filopodia formation, cell migration, endosome dynamics, and Golgi trafficking [45–56]. Additionally, RhoD appears to participate in other processes that affect cell proliferation, such as cell cycle regulation and cytokinesis [49,57]. Since JMY is involved in proteostasis and genotoxic stress responses through its activities in autophagy and apoptosis, we reasoned that the increase in *RHOD* expression in JMY^{KO} cells might be part of a compensatory pro-survival mechanism that allows the cells to better tolerate the loss-of-function mutation in *JMY*. To explore this possibility, we targeted RhoD for depletion using two independent siRNAs in JMY^{KO} cells. RT-PCRs verified that each RhoD siRNA reduced *RHOD* transcript levels (S11 Fig). DMSO- or etoposide-treated JMY^{KO} cells were then subjected to AnnV, PI, and Hoechst staining for measuring apoptosis and for evaluating necrosis (S11 Fig). Compared to control siRNA-treated samples, *RHOD*-depleted samples that were exposed to just DMSO showed modest increases in the fraction of cells with AnnV staining (S11 Fig), consistent with the possibility that cells are slightly more prone to apoptosis when *RHOD* levels are decreased. For the samples that were treated with etoposide, the low frequency of AnnV staining of JMY^{KO} cells was equivalent whether or not *RHOD* was silenced (S11 Fig). Since diminishing *RHOD* expression in these experiments did not restore apoptosis to normal levels, the intrinsic apoptotic defects in JMY^{KO} cells must not be due simply to the upregulation of *RHOD*. Perhaps more importantly, the percentage of AnnV-negative cells that were PI-positive, or considered to be

necrotic, was significantly higher when *RHOD* was depleted (S11 Fig). This latter result supports the idea that RhoD normally promotes cell survival.

Because of the complexities in studying RhoD activities during apoptosis in JMY$^{KO}$ cells, which appear to be fundamentally defective in their intrinsic apoptotic responses, we next wanted to characterize the effects of transiently depleting or permanently inactivating *RHOD* in otherwise healthy JMY-proficient cells. While *RHOD* transcript is below the limit of detection in the HAP1 cell line, it is expressed at measurable levels in eHAP cells (Fig 7A). So eHAP samples were treated with siRNAs for depleting *RHOD* (Fig 7B), exposed to DMSO or etoposide, and stained with AnnV and PI to assess the proportion of apoptotic and necrotic cells (Fig 7C). For the DMSO-treated populations, apoptotic death rose from 1% in cells receiving the control siRNA to 5% in cells receiving either of the RhoD siRNAs (Fig 7D), suggesting that RhoD depletion in healthy cells leads to a higher incidence of death under normal growth conditions. For the etoposide-treated samples, *RHOD* silencing caused apoptosis to occur in approximately 80% of eHAP cells, frequencies which were approximately double what was observed in cells treated with a negative control siRNA (Fig 7C and 7D). Moreover, the amount of RhoD transcript was inversely correlated with the percentage of AnnV-positive cells (Fig 7E). Necrotic death frequencies also rose, in this case from 2% in the presence of *RHOD* to 5% when it was silenced (Fig 7D). Thus, decreasing *RHOD* expression can increase stress-induced apoptosis and necrosis when normal JMY and p53 responses are present, further strengthening the conclusion that RhoD plays a basic pro-survival role in cells.

To assess the impact of a permanent loss of *RHOD* on cell survival and death, we next derived two independent RhoD$^{KO}$ cell lines from parental eHAP cells (S3 Fig). RT-PCRs verified that wild type RhoD mRNA was absent in the mutant cells (Fig 7F). Similar to other knockout cells, each of the RhoD$^{KO}$ cell lines had low levels of DNA damage when exposed to DMSO, but more numerous DNA breaks when incubated with etoposide (S4 Fig). Consistent with the transient depletion findings in etoposide-treated samples, apoptotic and necrotic death were more common in RhoD$^{KO}$ cells than in eHAP cells (Figs 7G and S11). Additionally, in dose-response experiments, at each etoposide concentration the RhoD$^{KO}$ cell lines showed a death frequency that was approximately twice that of parental cells (Fig 7H). These studies reveal that the permanent loss of RhoD makes cells more susceptible to dying during normal culture conditions and also enhances their apoptotic responses following acute DNA damage.

Since JMY and WHAMM are necessary for an efficient pathway of cyto *c* release, initiator caspase cleavage, and executioner caspase activation, we tested if *RHOD* inactivation affected these aspects of intrinsic apoptosis. Following 3h and 6h of etoposide treatment, compared to the eHAP parental cell line, the RhoD$^{KO}$ samples had significantly higher percentages of cells with diffuse cyto *c* localization, greater levels of initiator caspase-9 processing, more cells undergoing cleavage of DEVD-containing substrates, and more cells staining positive for active caspase-3 (Figs 7I–7K and S11). Together, the above results show that knocking out *RHOD* has the opposite apoptotic phenotypes of knocking out *JMY* or *WHAMM*. Thus, while JMY and WHAMM are WASP-family proteins that have important pro-apoptotic roles in intrinsic pathways of cell death, such functions may be constrained by pro-survival or anti-apoptotic activities of the small G-protein RhoD.

## The Arp2/3 complex is required for efficient intrinsic apoptosis

The actions of Rho-family G-proteins and WASP-family nucleation factors converge on the activation of the Arp2/3 complex, which nucleates actin into filaments [58,59]. To better understand the contribution of the actin cytoskeleton in intrinsic apoptosis, we first tested the

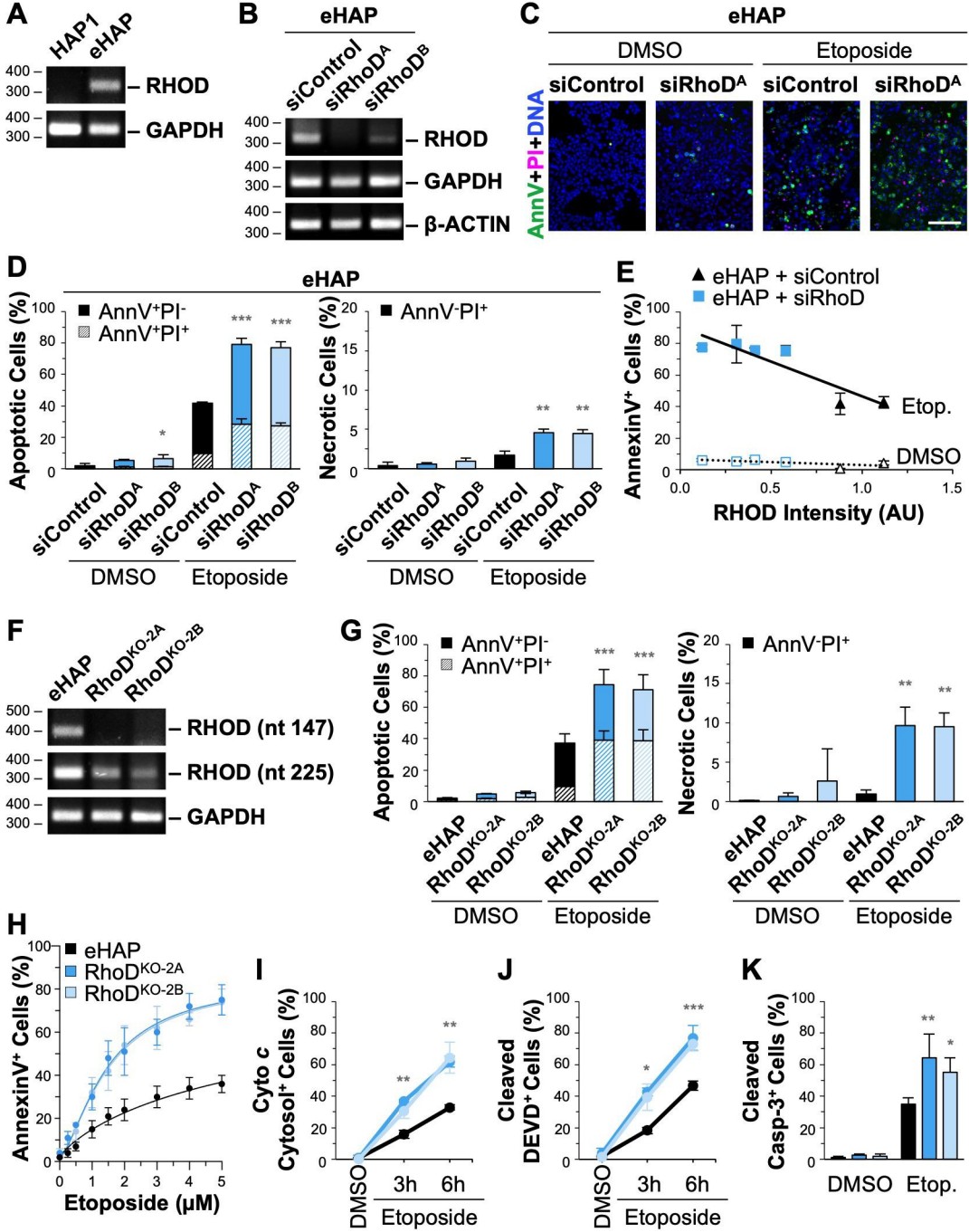

**Fig 7. Depletion or deletion of RhoD results in greater levels of apoptotic and necrotic cell death. (A)** RNA from HAP1 and eHAP cells was subjected to RT-PCR using primers to *RHOD* and *GAPDH* and visualized on agarose gels. **(B)** eHAP cells were treated with control siRNAs or independent siRNAs for the RHOD gene before performing RT-PCRs with primers to *RHOD*, *GAPDH*, and *β-ACTIN*. **(C)** Cells were treated with DMSO or 5μM etoposide for 6h and stained with Alexa488-AnnV (green), PI (magenta), and Hoechst (blue). Scale bar, 100μm. **(D)** The % of apoptotic cells was calculated and displayed as the fraction of AnnV-positive/PI-negative (AnnV⁺PI⁻) or AnnV/PI double-positive (AnnV⁺PI⁺) cells. Significance stars refer to comparisons of total AnnV⁺ counts for siControl vs siRhoD samples. The % of AnnV-negative/PI-positive (AnnV⁻PI⁺) necrotic cells was also quantified. Significance stars refer to comparisons of siControl vs siRhoD samples. Each bar represents the mean ±SD from 3 experiments (n = 3,662–4,948 cells per sample). **(E)** *RHOD* band intensities were normalized to *β-ACTIN* and plotted versus the % of AnnV-positive cells. Each point represents the mean ±SD from 3 fields-of-view in a given experiment (n = 980–1,728 cells per point). The slopes in the linear trendline regression equations for etoposide-treated samples (Y = -43.48X + 90.28) (p<0.001, R² = 0.75) and DMSO-treated samples (Y = -3.46X + 6.47) (p<0.01, R² = 0.37) were

significantly non-zero. **(F)** RNA from eHAP and RhoD$^{KO}$ cells was subjected to RT-PCR using primers to *RHOD* (forward primer: nt 124–147; reverse primer: nt 523–545), *RHOD* (forward primer: nt 225–247; reverse primer: nt 523–545), and *GAPDH*. Nucleotide 147 is predicted to be absent in mRNA from both knockouts, whereas nucleotide 247 should be present in both, albeit reflecting transcript levels that are lower than the parental control due to nonsense-mediated decay. Bands from this gel are also shown in panel (A). **(G)** eHAP and RhoD$^{KO}$ cells were treated and stained as in panel (D). The % of apoptotic and necrotic cells were quantified and each bar represents the mean ±SD from 3 experiments (n = 4,713–9,322 cells per sample). **(H)** eHAP and RhoD$^{KO}$ cells were treated with a range of etoposide concentrations and stained with Alexa488-AnnV and Hoechst. The % of AnnV-positive cells was calculated and each point represents the mean ±SD from 3–6 fields-of-view pooled from 1–2 experiments (n = 2,309–4,356 cells per experiment). Nonlinear regressions were performed with a baseline set to 0.02 and a maximum response set to 0.85. EC50 values were significantly different for parental vs KO samples (Mean EC50s: eHAP = 6.8μM; RhoD$^{KO}$ = 1.5μM, p<0.001). **(I)** Cells were treated with DMSO or etoposide, fixed, and stained with antibodies to detect cyto *c* and AIF. The % of cells with cytosolic cyto *c* staining was scored and each point represents the mean ±SD from 3 experiments (n = 480–843 cells per point). Significance stars refer to comparisons of parental to KO samples at the depicted timepoints. **(J)** Cells were stained with caspase-3/7 detection reagent, and the % of cleaved DEVD-positive cells was calculated. Each point represents the mean ±SD from 3 experiments (n = 1,746–3,580 cells per point). **(K)** Cells were treated with DMSO or etoposide for 6h before being fixed and stained with an antibody to cleaved caspase-3 and DAPI. The % of cleaved caspase-3-positive cells was calculated, and each bar represents the mean ±SD from 3 experiments (n = 1,227–7,830 cells per bar). Representative images and immunoblots appear in (S11 Fig). *p<0.05, **p<0.01, ***p<0.001 (ANOVA, Tukey post-hoc tests).

importance of the Arp2/3 complex by treating HAP1 and eHAP cells with siRNAs targeting two Arp2/3 subunits, *ACTR3* and *ARPC4*, whose depletion results in loss of the entire complex [37]. Similar to the knockdowns of p53, JMY, or WHAMM described earlier, Arp2/3 depletions resulted in significantly fewer AnnV-positive cells (Fig 8A and 8B) and significantly fewer cells exhibiting active cleaved caspase-3 staining (Fig 8C). Further, in HAP1 and eHAP cells, greater degrees of Arp2/3 complex knockdown resulted in larger reductions in apoptosis, and the amount of Arp3 protein was positively correlated with the percentage of AnnV-positive cells (Fig 8D). To validate the apparent requirement for the Arp2/3 complex in apoptosis, we also performed etoposide treatments in the presence of the small molecule Arp2/3 inhibitor, CK666. Consistent with the RNAi experiments, CK666-treated samples contained fewer active caspase-3-positive cells than did DMSO-treated control samples (Fig 8C).

To determine the effect of the Arp2/3 complex on apoptosis in the absence of JMY or WHAMM, we similarly performed Arp2/3 depletions in JMY$^{KO}$ and WHAMM$^{KO}$ cells (Fig 8A). Analogous to the observation that knockdown of p53 in JMY$^{KO}$ cells did not further diminish apoptosis levels from their already low amount of ~10% (Fig 5D–5F), the depletion of Arp2/3 in this knockout background did not affect AnnV or active caspase-3 staining frequencies (Fig 8B and 8C). In contrast, Arp2/3 silencing in WHAMM$^{KO}$ samples did result in significantly fewer apoptotic cells (Fig 8A–8C). One interpretation of these observations is that among the 20% of WHAMM-deficient cells that successfully execute apoptosis, approximately half rely on a death mechanism involving JMY and Arp2/3. Overall, it appears that the major intrinsic apoptotic pathway requires p53, JMY, and the Arp2/3 complex, and can be enhanced by WHAMM.

## JMY-driven actin assembly is associated with cyto *c* clustering and caspase-3 activation

JMY activates the Arp2/3 complex using a C-terminal domain consisting of 3 actin-binding WASP-homology-2 (W) motifs and an Arp2/3-binding Connector-Acidic (CA) motif, and can also use its WWW segment to nucleate actin directly [60]. To investigate whether the actin polymerization capacity of JMY is important for its participation in apoptosis, we generated 3 GFP-tagged JMY derivatives for performing rescue experiments in JMY$^{KO}$ cells: a wild type JMY construct (JMY$^{WT}$); a JMY mutant missing all 3 W motifs (JMY$^{ΔWWW}$) that can neither nucleate actin directly nor cooperate with the Arp2/3 complex; and a JMY mutant lacking the

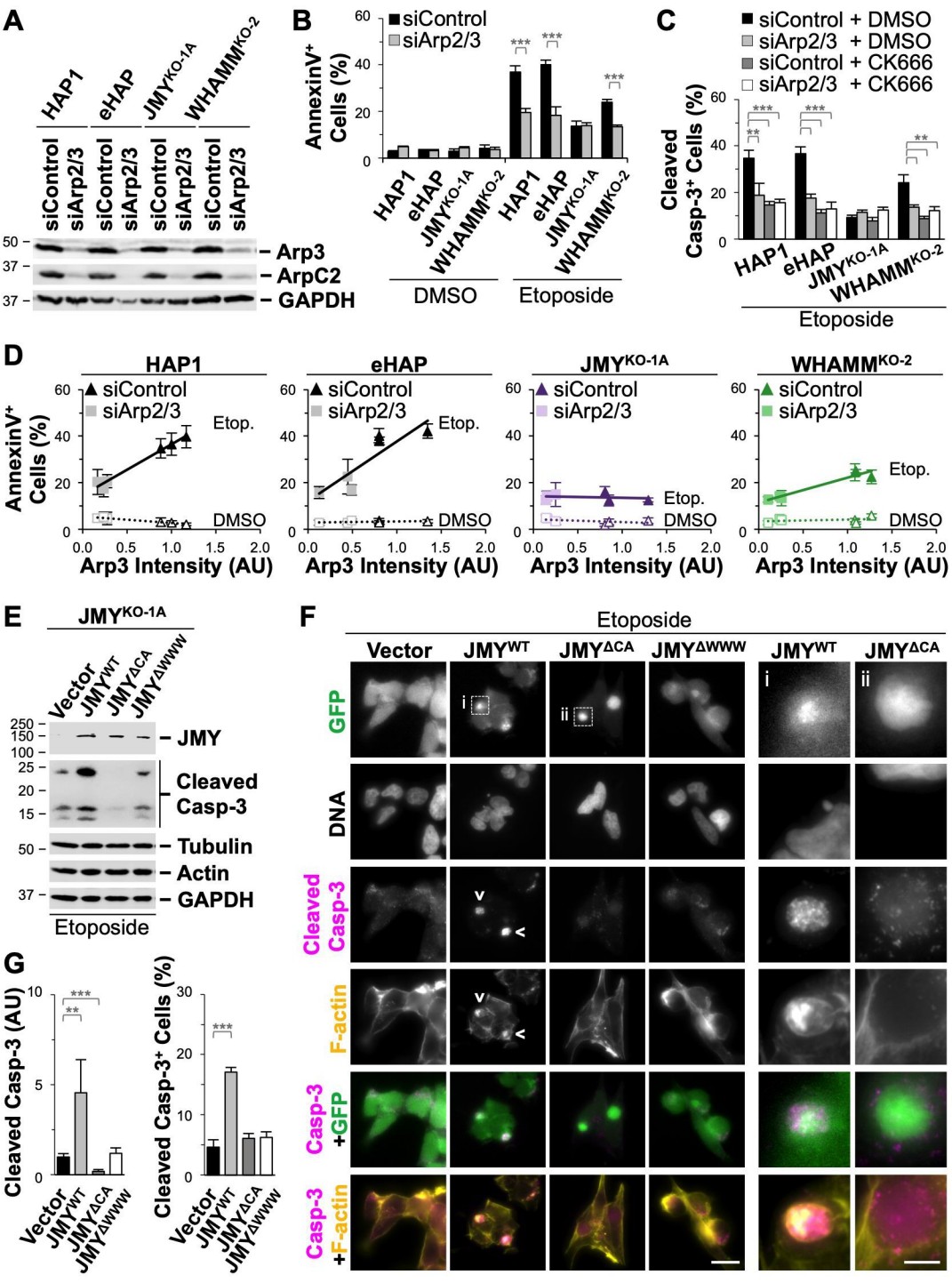

**Fig 8. Intrinsic apoptosis and caspase-3 activation require the Arp2/3 complex and actin assembly motifs in JMY. (A)** HAP1, eHAP, JMY[KO], and WHAMM[KO] cells were treated with control siRNAs or with siRNAs for the ACTR3 and ARPC4 genes before immunoblotting with antibodies to Arp3, ArpC2, and GAPDH. **(B)** Cells were treated with DMSO or 5μM etoposide for 6h and stained with Alexa488-AnnV, PI, and Hoechst. The % of AnnV-positive cells was calculated and each bar represents the mean ±SD from 3 experiments (n = 3,044–7,286 cells per bar). **(C)** Cells were treated with etoposide in conjunction with DMSO or CK666 before being fixed and stained with an antibody that recognizes active caspase-3 cleaved at Asp175 and DAPI. The % of cleaved caspase-3-positive cells was calculated and each bar represents the mean ±SD from 3 experiments (n = 2,532–4,651 cells per bar). **(D)** Arp3 band intensities on immunoblots were normalized to GAPDH intensities and plotted against the % of AnnV-positive cells. Each point represents the mean ±SD from 3 fields-of-view in a

given experiment (n = 974–2,499 cells per point). The slopes in the linear trendline regression equations for HAP1, eHAP, and WHAMM$^{KO}$ etoposide-treated samples (HAP1: Y = 22.43X + 14.39; eHAP: Y = 25.26X + 12.411; WHAMM$^{KO}$: Y = 10.901X + 11.31) were significantly non-zero (p<0.001, HAP1: $R^2$ = 0.85; eHAP: $R^2$ = 0.71; WHAMM$^{KO}$: $R^2$ = 0.82). AU = Arbitrary Units. **(E)** JMY$^{KO}$ cells transfected with plasmids encoding GFP (vector) or GFP-tagged JMY constructs (JMY$^{WT}$, JMY$^{\Delta CA}$, JMY$^{\Delta WWW}$) were treated with 10μM etoposide for 5h before immunoblotting with antibodies to JMY, active cleaved caspase-3, tubulin, actin, and GAPDH. **(F)** GFP-expressing cells (green) treated as in (E) were also fixed and stained with DAPI to detect DNA, an antibody to visualize cleaved caspase-3 (magenta), and phalloidin to label F-actin (yellow). Arrowheads highlight examples of cleaved caspase-3-positive clusters. Magnifications represent examples of a JMY$^{WT}$ cluster that overlaps with a cleaved caspase-3 cluster and is associated with an F-actin-rich territory (i), and a JMY$^{\Delta CA}$ structure lacking both cleaved caspase-3 and F-actin enrichment (ii). Scale bars, 25μm, 10μm. **(G)** For quantification of (E), the cleaved caspase-3 band intensities were normalized to the loading control intensities and the vector sample was set to 1. Each bar represents the mean intensity ±SD from 4 experiments. AU = Arbitrary Units. For quantification of (F), the % of cleaved caspase-3-positive cells was calculated and each bar represents the mean ±SD from 3 experiments (n = 3,447–6,084 cells per bar). $^{**}$p<0.01, $^{***}$p<0.001 (t-tests).

CA segment (JMY$^{\Delta CA}$) that cannot activate Arp2/3 but retains its actin nucleating WWW region. The GFP-JMY plasmids, or a vector control, were introduced into the JMY$^{KO-1A}$ line, which was then cultured in selective media, exposed to etoposide, and subjected to immuno-blotting (Fig 8E) or immunostaining (Fig 8F) for active cleaved caspase-3. Compared to the vector-transfected knockout cells, which contained low levels of cleaved caspase-3, cells transfected with the wild type JMY construct showed a 4.5-fold increase in the amount of cleaved caspase-3 and a 3.5-fold increase in the cellular frequency of cleaved caspase-3 staining (Fig 8G), indicating that full-length JMY can restore intrinsic apoptosis functions to JMY$^{KO}$ cells. In contrast, the JMY mutant with the WWW segment deleted failed to rescue apoptosis in the knockout cells, as cleaved caspase-3 levels were indistinguishable from those of the vector-transfected controls (Fig 8E and 8G). The JMY mutant lacking the CA portion was also inef-fective at increasing the frequency of cleaved caspase-3 staining (Fig 8G), and actually decreased the total levels of active caspase-3 in JMY$^{KO}$ cells (Fig 8E and 8G). These results show that JMY requires its actin assembly activity, particularly its Arp2/3 complex binding segment, in order to drive intrinsic apoptosis.

JMY is primarily cytoplasmic in healthy cells [27,28], and is typically redistributed into both the cytoplasm and nucleus upon exposure to apoptosis-inducing stressors [29,30]. Con-sistent with this expectation, GFP-JMY exhibited a fairly uniform localization in the cytosol and nucleus (Fig 8F). Interestingly, however, GFP-JMY additionally appeared in intense juxta-nuclear clusters (Fig 8F), and those cytoplasmic structures overlapped with clusters of cleaved caspase-3 (Fig 8F). Further, staining with fluorescent phalloidin demonstrated that filamen-tous- (F-) actin also assembled throughout the caspase-3 region and was most intense adjacent to JMY (Fig 8F). JMY$^{\Delta CA}$ and JMY$^{\Delta WWW}$ could also be found in bright cytoplasmic spots, but cleaved caspase-3 staining was diffuse within these areas and F-actin was not enriched (Fig 8F). These findings highlight important relationships between JMY-mediated actin polymeri-zation and the formation of cytoplasmic territories containing active caspase-3.

Because caspase-3 activation is driven by apoptosomes, macromolecular platforms consist-ing of cyto *c*, procaspase-9, and other scaffolding factors [38,61,62], we revisited the potential connections among JMY, cyto *c*, and actin in the cytoplasm. Reminiscent of the GFP-JMY and cleaved caspase-3 clusters, punctate juxtanuclear structures of endogenous JMY were visible in a subset of etoposide-treated HAP1 cells (Fig 9A and 9B). The same type of staining pattern was also observed in 15–20% of U2OS cells, which are larger and easier to image (Fig 9A and 9B). Interestingly, in both HAP1 and U2OS cells, many of these JMY puncta colocalized with cyto *c* puncta that were distinct from mitochondria (Figs 9C, 9D, and S12). They were also associated with bright clouds of the cleaved DEVD reporter (prior to its nuclear import), indic-ative of active caspase-3 being present in proximity to these JMY- and cyto *c*-containing

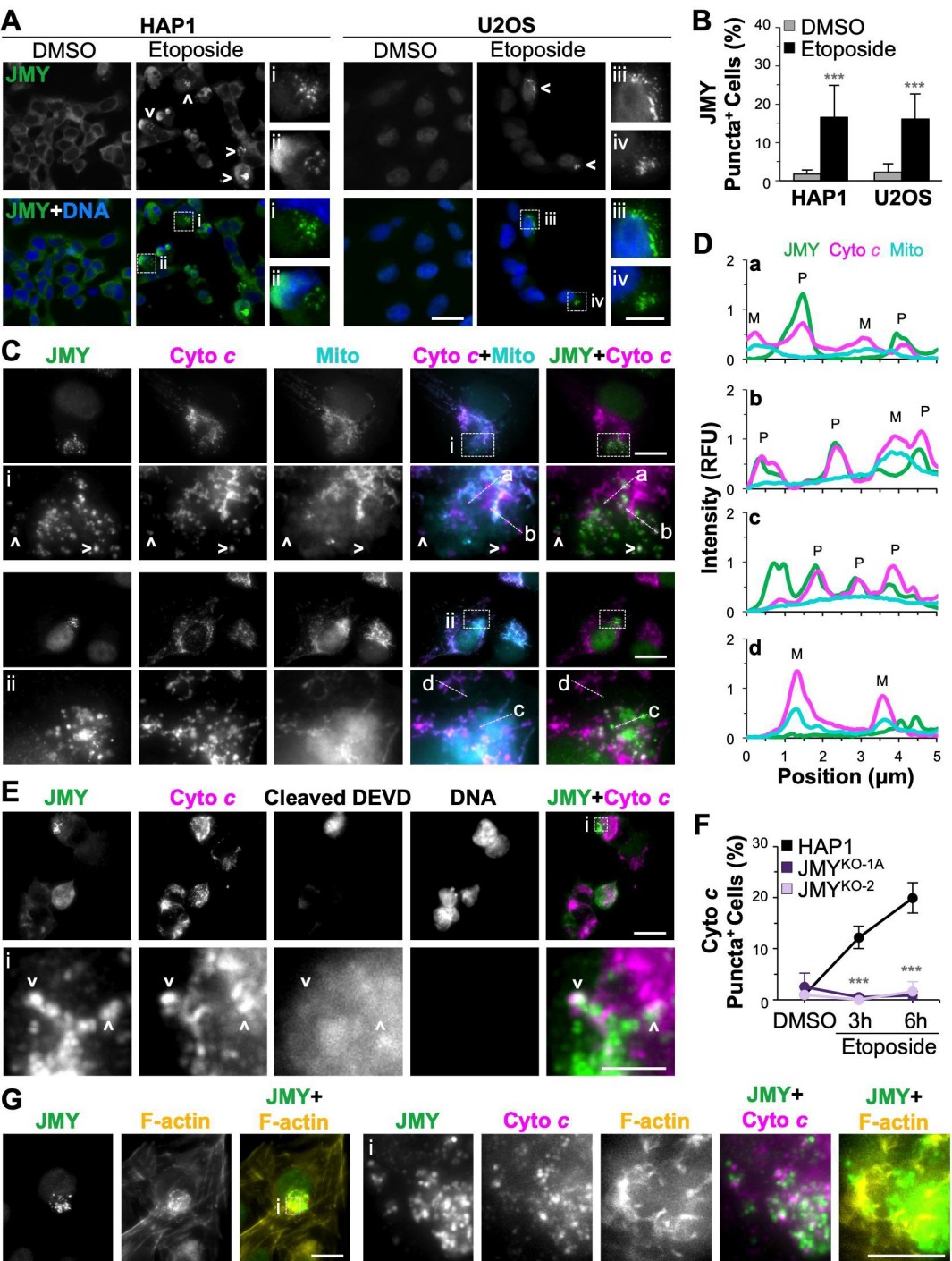

**Fig 9. JMY colocalizes with cytochrome *c* puncta and is required for their assembly into F-actin-associated cytoplasmic clusters. (A)** HAP1 and U2OS cells were treated with DMSO, 5μM etoposide, or 10μM etoposide for 6h before being fixed and stained with an antibody to JMY (green) and DAPI (DNA; blue). Arrowheads and magnifications (i-iv) highlight examples of cytosolic JMY puncta. Scale bars, 25μm, 10μm. **(B)** The % of cells with JMY puncta was calculated in ImageJ by counting the cells containing JMY puncta and dividing by the total number of DAPI-stained nuclei. Each bar represents the mean % ±SD from 3 experiments (n = 1,149–1,576 HAP1 cells per bar; n = 466–489 U2OS cells per bar). Significance stars refer to comparisons between DMSO and etoposide treatments. **(C)** U2OS cells were treated with 10μM etoposide for 6h and incubated with MitoTracker (Mito; cyan) before being fixed and stained with antibodies to JMY (green) and cyto *c* (magenta). Arrowheads and magnifications (i,ii) show examples of cytosolic JMY and cyto *c* puncta. Scale bars, 10μm. **(D)** 5μm lines were drawn through the magnified images in (C) using ImageJ to measure the pixel intensity profiles for JMY, cyto *c*, and

mitochondria, highlighting the differences in cyto *c* maintained within mitochondria (M) and cytosolic cyto *c* that colocalizes with JMY puncta (P). **(E)** HAP1 cells were treated with 5μM etoposide for 6h, stained with caspase-3/7 green detection reagent to label cleaved DEVD, fixed, and stained with antibodies to JMY (green), cyto *c* (magenta), and DAPI. Scale bars, 25μm, 10μm. Magnifications represent examples of JMY and cyto *c* puncta that overlap with an area containing cleaved DEVD. **(F)** HAP1 and JMY$^{KO}$ cells were treated with DMSO or etoposide before being fixed and stained to visualize AIF and cyto *c*. The % of cells with non-mitochondrial cyto *c* puncta was calculated in ImageJ. Each point represents the mean % ±SD from 3 experiments (n = 194–332 cells per point). Significance stars refer to comparisons of parental to KO samples at the depicted timepoints. **(G)** U2OS cells were treated with etoposide, fixed, and stained with antibodies to JMY (green) and cyto *c* (magenta), and with phalloidin (F-actin; yellow). Magnifications (i) depict clusters of JMY, cyto *c*, and F-actin. Scale bars, 25μm, 10μm. ***p<0.001 (ANOVA, Tukey post-hoc tests).

clusters (Fig 9E). To test whether JMY was required for cells to form the distinct cytoplasmic cyto *c* structures, we quantified the proportion of HAP1 and JMY$^{KO}$ cells containing mito-chondria-independent cyto *c* puncta. While the fraction of parental cells harboring cyto *c* puncta increased during 3h and 6h exposures to etoposide, the JMY$^{KO}$ lines did not form cyto-solic cyto *c* puncta under any of these conditions (Fig 9F). In addition, inspection of U2OS cells revealed that the overlap of JMY and cyto *c* puncta took place within F-actin-rich territo-ries (Figs 9G and S12). F-actin appeared to be specifically reorganized, rather than globally polymerized or upregulated, as the inactivation of *JMY* did not significantly affect the overall levels of actin filaments, stained with phalloidin, or the total amounts of actin in the cytoplasm or nucleus, stained with an actin antibody (S12 Fig). These observations are consistent with a role for cytoplasmic JMY and F-actin during the period of apoptosis encompassing apopto-some assembly and executioner caspase-3 activation.

Finally, we investigated whether the formation of the cytosolic JMY puncta was impacted by WHAMM or RhoD. After etoposide treatment, 20% of HAP1 and eHAP cells contained clusters of JMY puncta while only 10% of WHAMM$^{KO}$ cells but more than 35% of RhoD$^{KO}$ cells displayed such cytosolic JMY puncta (Fig 10A and 10B). These phenotypic differences are reminiscent of the reduced versus increased apoptotic characteristics previously observed in the WHAMM and RhoD knockout cells, respectively. To address how overexpression of RhoD might influence the formation and organization of the JMY-containing structures, we tran-siently transfected U2OS cells with constructs encoding GFP-tagged wild type RhoD (RhoD$^{WT}$) or a GDP-locked dominant negative mutant RhoD (RhoD$^{DN}$) (Fig 10C). JMY puncta were formed at indistinguishable frequencies in cells expressing either GFP-RhoD$^{WT}$ or the GFP vec-tor control (Fig 10D), suggesting that elevating RhoD protein levels does not inherently disturb this pattern of JMY localization during apoptotic signaling. In contrast, only half as many cells expressing the RhoD$^{DN}$ mutant contained JMY puncta (Fig 10D). Additionally, this dominant interfering RhoD protein increased JMY levels in the nucleus (Fig 10C–10E), signifying that the nucleotide-bound state of RhoD affects both the formation of the punctate cytosolic structures and the nuclear accumulation of JMY. Further examination of GFP-RhoD localization and F-actin staining revealed that RhoD$^{WT}$ surrounded the clusters of JMY puncta and allowed actin polymerization (Fig 10C and 10E). Conversely, RhoD$^{DN}$ did not localize in analogous juxtanuc-lear regions or permit actin filaments to form there (Fig 10C and 10E), suggesting that RhoD-GDP can act as an 'off switch' for JMY-mediated cytoskeletal rearrangements. Collec-tively, these results indicate that although JMY and RhoD exhibit opposing pro-apoptotic versus pro-survival activities, both proteins apparently function in the assembly and organization of cytoplasmic territories containing cyto *c*, active caspases, and F-actin.

## Discussion

The polymerization, organization, and turnover of actin filaments have been thoroughly char-acterized during many cellular functions that maintain viability [9,10]. However, the

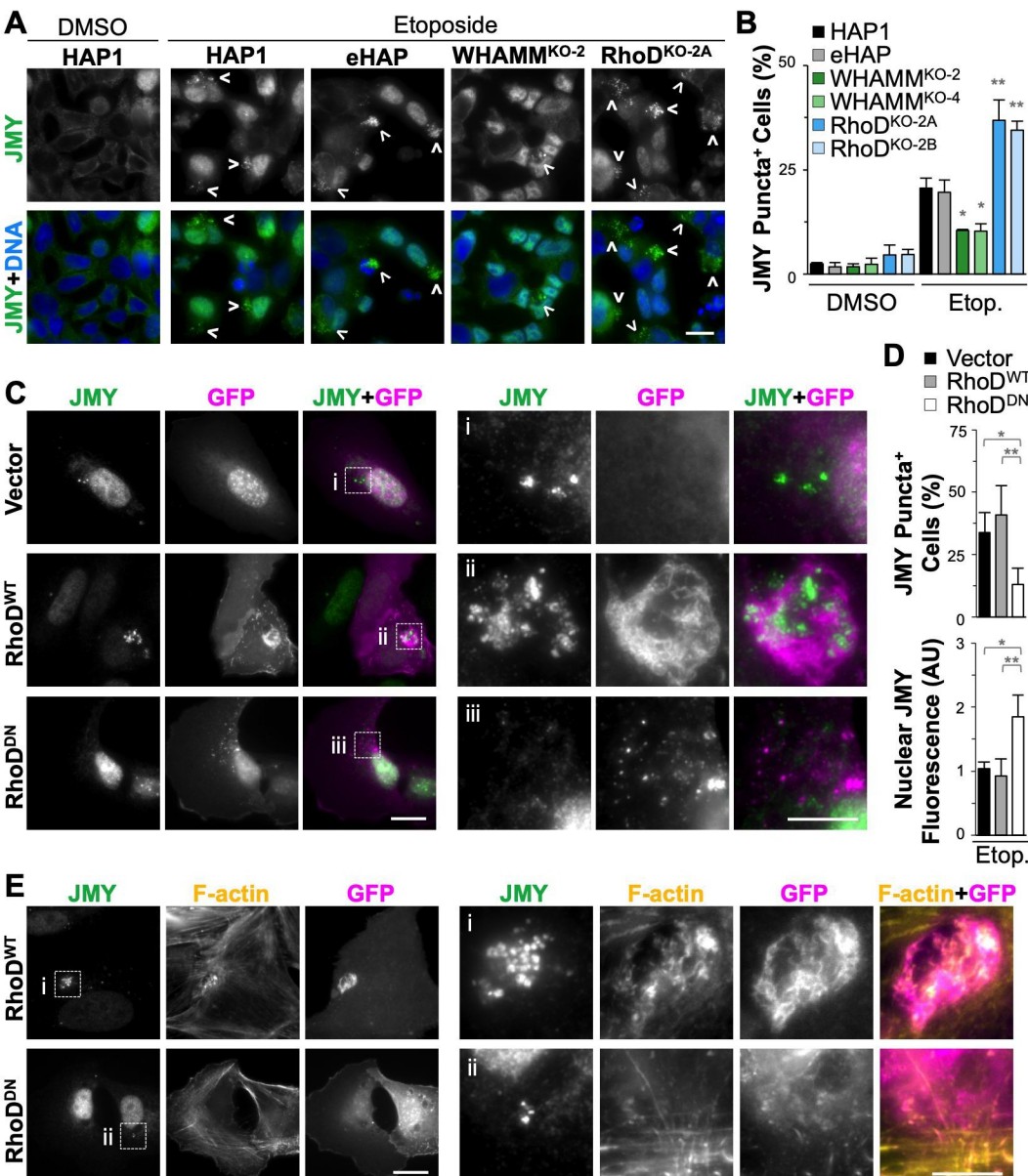

**Fig 10. WHAMM and RhoD affect the organization of cytoplasmic territories containing JMY, cytochrome *c*, and F-actin. (A)** HAP1, eHAP, WHAMM[KO], and RhoD[KO] cells were treated with DMSO or 5μM etoposide for 6h before being fixed and stained with a JMY antibody (green) and DAPI (DNA; blue). Arrowheads highlight examples of cytosolic JMY puncta. Scale bar, 25μm. **(B)** The % of cells with JMY puncta was calculated using ImageJ. Each bar represents the mean % ±SD from 2 experiments (n = 170–346 cells per bar). Significance stars refer to comparisons to parental cell lines. **(C)** U2OS cells transfected with plasmids encoding GFP (vector) or GFP-RhoD constructs (RhoD[WT] and RhoD[DN]) were treated with 10μM etoposide for 6h before fixing and staining with a JMY antibody (green). GFP is shown in magenta. Scale bars, 25μm, 10μm. Magnifications (i-iii) represent examples of JMY localization. **(D)** The % of U2OS cells with JMY puncta was calculated in ImageJ and each bar represents the mean % ±SD from 3–4 experiments (n = 107–201 cells per bar). U2OS cells treated as in (C) were also stained with DAPI. Nuclear JMY fluorescence intensities were measured in ImageJ relative to those of the vector sample. Each bar represents the mean ±SD from 3 experiments (n = 105–186 cells per bar). AU = Arbitrary Units. **(E)** U2OS cells treated as in (C) were stained with phalloidin to label F-actin (yellow). Magnifications (i,ii) represent examples of GFP-RhoD localizations. Scale bars, 25μm, 10μm. *p<0.05, **p<0.01 (ANOVA, Tukey post-hoc tests).

contributions of actin assembly factors to apoptotic cell death are not well understood. Among the best-known cytoskeletal features of apoptosis are the actin filament rearrangements and disassembly that accompany changes in cell adherence and morphology [7,8]. Actin itself can be cleaved by caspases [63,64], resulting in non-polymerizable fragments that promote apoptotic cellular phenotypes [65–67]. The actin turnover machinery has also been implicated in controlling apoptosis at multiple steps. Cofilin, which depolymerizes actin filaments [68], influences the early stages of apoptosis, as it translocates to mitochondria and may increase the release of cytochrome *c* [69–71] and the localization of p53 [72]. Actin is additionally recruited to mitochondria at around the time of mitochondrial permeabilization [73–75]. During the later stages of apoptosis, the F-actin-severing protein gelsolin [76] is cleaved by caspases, resulting in a fragment with unregulated severing activity that drives depolymerization [77] and allows chromatin fragmentation [78]. In contrast to these described roles for actin disassembly proteins in apoptosis, less is known about the actions of the actin assembly machinery during cell death. Our results reveal intrinsic apoptotic functions for JMY, WHAMM, the Arp2/3 complex, and RhoD, and create a framework for understanding how actin nucleation factors and small G-proteins control pro-apoptotic and pro-survival processes.

WASP-family proteins have many well-recognized activities, but the extent to which they participate in cell death pathways is relatively underexplored. WAVE1 has been one of the most studied members, as it can affect the localization or modification of Bcl-2-family proteins. Several transient overexpression and depletion experiments suggest that WAVE1 limits mitochondrial permeabilization [20–23]. But in our initial knockout screen using BCR-ABL-immortalized human cells, the permanent loss of WAVE1 did not cause a consistent increase in the frequency of apoptosis, so the degree to which WAVE1 acts as an anti-apoptotic factor in a cell type specific manner requires further investigation.

In addition to WAVE1, JMY has been studied in the context of apoptosis. JMY was discovered as a cofactor that enhanced the transcriptional activity of p53 [24], and subsequent RNAi studies indicated that JMY has pro-survival [32] or pro-death [27,29,79] functions under different experimental circumstances. Our work advances the understanding of its pro-death role by demonstrating that cells with permanent deletions or transient depletions of JMY exhibit significant defects in apoptotic processes following DNA damage. In fact, inactivation of each WASP-family gene revealed that only cells lacking JMY, or its closest homolog WHAMM, had substantial deficiencies in their intrinsic apoptotic responses. WHAMM did not have any previously described roles in apoptosis, so the latter observation adds a third WASP-family member to the repertoire of actin assembly factors that can function in programmed cell death.

WHAMM and JMY comprise a subgroup within the WASP family, are approximately 50% similar to one another, and both promote anterograde membrane transport [37,47,80] and autophagy [32,81,82]. Although the two proteins participate in similar cellular activities, they exhibit some key molecular differences. WHAMM has a conventional C-terminal WWCA domain that activates Arp2/3 complex-mediated actin assembly [37], whereas JMY has an extended WWWCA domain wherein the WCA produces Arp2/3-branched networks, and the WWW directly nucleates linear filaments [60]. WHAMM localizes to the ER and *cis*-Golgi and binds microtubules to mediate membrane tubulation and transport [37,47,83], while JMY acts later in the secretory pathway to promote trafficking away from the *trans*-Golgi [80]. WHAMM and JMY also both function in multiple aspects of autophagy. WHAMM binds to the phospholipid PI(3)P and stimulates the Arp2/3 complex to accelerate lipidation of LC3-family proteins during autophagosome biogenesis [82]; WHAMM additionally drives Arp2/3-dependent autophagosome movement [81] and autolysosome remodeling [84]. JMY binds to LC3 and polymerizes actin at autophagosomes during their maturation [32] and actin-based motility [28]. While it is reasonable to assume that the functions of WHAMM and

JMY in secretion and autophagy help sustain cells during normal growth and allow them to adapt to conditions of nutritional stress, the fact that both factors also enable cell death indicates that they function as pivotal players in the cellular responses to multiple other stressors including genotoxic insults.

Following etoposide-induced DNA damage, parental HAP1 cells executed an immediate p53-dependent cell death program. In contrast, JMY or WHAMM knockout cells showed significant reductions in the terminal apoptosis phenotypes of phosphatidylserine externalization and membrane permeabilization. Further comparisons of JMY, WHAMM, and WHAMM/JMY knockout cells revealed that an individual WHAMM deficiency results in partial apoptotic deficits, whereas JMY ablation, either alone or in conjunction with WHAMM, causes a more severe defect in apoptosis–one equivalent to a loss of p53.

Importantly, cells lacking JMY or WHAMM displayed inefficiencies and/or kinetic delays in multiple apoptotic processes, including the mitochondrial export of cyto *c*, caspase activation, and effector caspase cleavage of target proteins. These results indicate that both cytoskeletal regulators are key participants in a rapid, intrinsic, p53-dependent cell death pathway, with JMY acting as the more prominent contributor. They further suggest that JMY and WHAMM function at one or more steps in the apoptotic pathway that take place after the recognition of DNA damage and arrest of cell proliferation, including during transcriptional reprogramming, mitochondrial permeabilization, apoptosome assembly, and caspase cleavage/activation.

Under normal cell culture conditions, JMY is found predominantly in the cytosol [27,28], but in response to DNA damage, it accumulates in the nucleus while still maintaining a cytosolic presence [29,30]. At steady-state, WHAMM is mostly associated with cytoplasmic membrane-bound organelles, although it displays a nuclear localization in a small subpopulation of cells [37]. Such observations are consistent with the existence of both cytoplasmic and nuclear roles for these proteins in apoptosis. JMY was previously shown to interact with the stress-response protein Strap, the acetyltransferase p300, and p53 [24,31]. p300 is one of many cofactors known to modify p53 to promote the transcription of genes necessary for initiating cell cycle arrest and stimulating pro-death pathways in response to cell damage [26,85–87]. While the precise function of JMY in a complex with Strap, p300, and p53 is unknown, co-overexpression of JMY with p53, or with p53 plus p300, augments *BAX* transcription in several epithelial cell contexts [24,29,79], and introduction of a JMY siRNA into p53-overexpressing cells limits *BAX* induction [29]. However, our current work suggests that the activity of JMY in transcriptional regulation via p53 may not be the primary apoptotic driver following DNA damage.

p53 shuttles between the cytoplasm and nucleus, and although it has pro-apoptotic functions in the cytosol [88,89], its most extensively characterized activities are as a nuclear transcription factor [43,90,91]. In our experiments, etoposide-induced increases in the overall abundance, nuclear accumulation, phosphorylation, and acetylation of p53 occurred in the presence or absence of JMY. Moreover, *JMY* inactivation had little effect on transcriptional upregulation of the p53 target *CDKN1A*, which encodes the key cell cycle inhibitor p21. In fact, JMY-deficient cells exhibited a prolonged arrest in response to DNA damage, indicating that JMY is not necessary for the gene expression changes that stop proliferation. These results collectively support the idea that JMY participates in only a subset of the p53-mediated transcriptional responses to genotoxic stress.

Given the speed and efficiency with which p53- and JMY-proficient parental cells underwent apoptosis, we hypothesized that they constitutively express a pool of potentially apoptotic factors that enable the cells to respond rapidly to DNA damage. Indeed, global and targeted gene expression profiling experiments implied that inactivation of *JMY* did not adversely affect the expression of common apoptosis regulators either before or after etoposide exposure.

However, it was striking that the JMY knockout cells experienced a significant upregulation of *RHOD*, which encodes a small G-protein previously shown to interact with WHAMM [45,46]. RhoD has wide-ranging functions in regulating actin dynamics at the plasma membrane, endosomes, and Golgi, and in influencing cell proliferation [45–57]. While the selective pressures that led to greater *RHOD* expression in two out of the three JMY knockout cell lines are unknown, we nevertheless reasoned that increased RhoD levels may be part of a compensatory pro-survival mechanism that allows cells to better tolerate a permanent *JMY* mutation. Consistent with this possibility, RhoD depletion or deletion experiments resulted in more necrosis and enhanced apoptotic responses–the opposite phenotype of JMY- or WHAMM-depleted cells. These findings reveal complex relationships among two actin nucleation factors and one small G-protein during cell death and survival (S12 Fig).

Given that the best understood molecular behaviors of JMY and WHAMM reside in their abilities to promote actin assembly in the cell cytoplasm, we tested whether the Arp2/3 complex was also important for the execution of apoptosis. Indeed, RNAi-mediated depletion or pharmacological inhibition of the Arp2/3 complex also impaired apoptosis, and JMY truncations that cannot activate Arp2/3 failed to increase the apoptotic capacity of JMY knockout cells. Most notably, wild type and actin polymerization-deficient mutants of JMY were both able to congregate into discrete cytosolic structures, but only the polymerization-proficient protein generated distinct juxtanuclear zones containing actin filaments, active caspase-3, and cytochrome *c*. In light of these observations, we propose that JMY-mediated actin assembly is required for creating cytoplasmic territories that direct the activation of executioner caspase-3 by cytochrome *c*-containing apoptosomes (S12 Fig). The structural and biophysical properties of apoptosomes have begun to come into focus [38,61,92], yet their cell biological characteristics remain less clear. Since JMY clusters form less frequently in WHAMM[KO] cells and with different frequencies depending on RhoD and its nucleotide-bound state, it will be important to gain a better understanding of the physical connections among RhoD, WHAMM, JMY, Arp2/3, and actin, and how they impact the organization and function of apoptosomes and caspases.

Continuing to define the roles of different elements of the actin assembly and regulatory machinery in cellular adaptations to endogenous or exogenous stress could provide new avenues for understanding tumorigenesis. The Arp2/3 complex and WASP-family members influence cell motility [13,14], and elevated levels of nucleation factors are associated with increased metastasis [93,94], indicating that such actin cytoskeletal proteins have proto-oncogenic features. Similarly, the functions of WHAMM and JMY in autophagy [32,81,82] could enable cancer cells to better survive in diverse physiological environments and in response to chemotherapy [95,96]. However, our current results describing apoptotic requirements for JMY and WHAMM support the idea that these factors also possess key tumor-suppressive features. *TP53* is well known as the most commonly mutated gene in human cancers, and mutations that inactivate p53 or other p53-associated proteins are linked to poor prognoses [97–99]. Interestingly, JMY expression also appears to be lost in several B-cell lymphomas and invasive carcinomas [100]. Given the new positions of JMY and WHAMM as important players in p53-dependent apoptotic pathways, a greater understanding of how their actin nucleation, proto-oncogenic, and tumor-suppressive activities are coordinated will likely shed further light on how programmed cell death mechanisms are impacted by the cytoskeleton.

## Materials and methods

### Ethics statement

Research with biological materials in the Campellone Lab was approved by the UConn Institutional Biosafety Committee. This study did not include research with human subjects or live

animals. Human cell lines were acquired from Horizon Genomics or the UC Berkeley cell culture facility as described below.

## Biological materials

Cell lines are listed in S1 Table, nucleic acids are listed in S2 Table, and key reagents are listed in S3 Table. HAP1 cells are nearly-haploid fibroblast-like cells that contain an immortalizing BCR-ABL fusion and a single copy of all chromosomes except for a heterozygous 30Mb fragment of chromosome 15, which is integrated within the long arm of chromosome 19 [33,101]. This diploid portion encompasses 330 genes, including *WHAMM*. CRISPR/Cas9-engineered eHAP cells were derived from the HAP1 line and are fully haploid [35]. In the current study, CRISPR/Cas9-mediated recombination using guide RNAs against target genes resulted in frameshift and/or splicing mutations (S1 Table and S3 Fig) which were confirmed by DNA sequencing (Horizon Genomics). HAP1 cells were used for mutagenesis of *JMY*, *WASL* (encoding N-WASP), *WASF1* (WAVE1), *WASF2* (WAVE2), *WASF3* (WAVE3), *BRK1* (WAVE Complex), *CCDC53* (WASH Complex), and *CTTN* (Cortactin), while eHAP cells were used for mutagenesis of *WHAMM* and *RHOD*. The latter two cell lines were made in an eHAP background due to the *WHAMM* diploidy in HAP1 cells, and because under normal culture conditions *RHOD* transcript was detectable in eHAP but not HAP1 cells (Horizon Genomics). One of the *WHAMM* mutant cell lines (WHAMM$^{KO-2}$) was studied previously [82], and was used to make the WHAMM/JMY$^{DKO}$ cell lines. Attempts to generate *WASH1* KO cells were unsuccessful, as all viable clones contained in-frame mutations (Horizon Genomics). HAP1 cell derivatives were cultured in Iscove's Modified Dulbecco's Medium (IMDM) supplemented with 10% fetal bovine serum (FBS) and penicillin-streptomycin. U2OS and HeLa cells (UC Berkeley cell culture facility) were cultured in Dulbecco's Modified Eagle's Medium (DMEM) supplemented with 10% FBS and antibiotic-antimycotic. All cell lines were grown at 37˚C in 5% $CO_2$. All assays were performed using cells that had been in active culture for 2–10 trypsinized passages.

## Transgene expression

For JMY cloning (S2 Table), DNA fragments were amplified from cDNA templates by PCR, digested, and ligated into the BamHI and NotI restriction sites of pKC425 [102]. JMY$^{WT}$ contains amino acids 1–983 of murine JMY, JMY$^{\Delta WWW}$ contains amino acids 1-856/934-983, and JMY$^{\Delta CA}$ contains amino acids 1–941. For RhoD cloning (S2 Table), a BamHI-EcoRI restriction fragment from GST-RhoD [45] was inserted into the BglII and EcoRI sites of pKC-LAP-C1 [37]. RhoD$^{WT}$ contains amino acids 1–210 of murine RhoD, and RhoD$^{DN}$ harbors a T31N mutation that was introduced by PCR-based site-directed mutagenesis. Plasmids were maintained in *E.coli* XL-1 Blue (Stratagene). For rescue studies, JMY$^{KO-1A}$ cells cultured in 12-well plates were transfected with 2μg of linearized plasmid encoding GFP or GFP-JMY derivatives using LipofectamineLTX (Invitrogen). 24h later, the cells were transferred to media containing 1.5mg/mL G418 for 12–16 days. Surviving colonies were collected, expanded in 24-well plates, 6-well plates, and ultimately T-25 flasks prior to cryopreservation. Upon re-animation, G418 concentrations were reduced stepwise to 1.0mg/mL, 0.5mg/mL, and 350μg/mL in 6-well plates. The cell populations were subjected to experimental manipulations within 5 passages in media containing 350μg/mL G418. For transient expression, U2OS cells were transfected in 6-well plates with 500ng of DNA prior to reseeding onto 12mm glass coverslips in 24-well plates and fixing as described below. Transfection increased cell death in Fig 10.

## Chemical treatments and RNA transfections

Cells were treated with different concentrations of etoposide (Sigma Aldrich) diluted from an initial stock of 10mM in DMSO or with 100μM CK666 (Millipore) from a 100mM stock. Equivalent volumes of DMSO were used as controls. For RNA-seq experiments, the control condition consisted of regular media, while the treated condition consisted of media in which the etoposide powder was dissolved directly to 5μM. For JMY rescue experiments, to be able to detect cleaved caspase-3 in the JMY^KO cell line, etoposide was dissolved to 10μM. For RNAi experiments, cells were grown in 6-well plates for 24h, transfected with 40nM siRNAs (S2 Table) using RNAiMAX (Invitrogen), incubated in growth media for 24h, reseeded into 6-well plates and 24-well glass-bottom plates (MatTek), and incubated for an additional 24h. Cells cultured in 6-well plates were collected and processed for immunoblotting or RT-PCR, and cells cultured in 24-well plates were used in live fluorescence microscopy assays.

## RT-PCR and RNA-sequencing

For RT-PCRs, approximately $10^6$ cells were seeded in 6-well plates, and for RNA-seq $2.5 \times 10^6$ cells were seeded in 6cm dishes. After 24h of growth, cells were subjected to control-, DMSO-, or etoposide-containing media changes and rinsed with phosphate buffered saline (PBS). RNA was isolated using TRIzol reagent (Ambion), followed by a chloroform extraction, isopropanol precipitation, and 75% ethanol wash before resuspending RNA in water. For RT-PCRs, RNA was reverse transcribed into cDNA using Superscript III RT (Invitrogen) and subsequently amplified using Taq polymerase (New England Biolabs) and gene-specific primers (S2 Table). Primers were designed to amplify ~230-460bp products from each cDNA template. After agarose gel electrophoresis, band intensities for *CDKN1A*, *WHAMM*, and *RHOD* were quantified using ImageJ software [103] and normalized to *β-ACTIN* and/or *GAPDH* bands for each sample. For RNA-seq, Illumina cDNA library preparation was based on the Illumina TruSeq Stranded mRNA sample preparation guide. Total reads from RNA-seq were aligned with TopHat, and differential expression was calculated with the CuffLinks/CuffDiff program. Gene expression values were given as Fragments Per Kilobase of transcript per Million mapped reads (FPKM), differential expression was calculated as the Log2(FPKM ratio JMY^KO-1A:HAP1), and data from 3 independent experiments were merged. When comparing the transcriptomic changes that occurred in HAP1 or JMY^KO-1A lines upon etoposide treatment, differential expression was calculated as the Log2(FPKM Etoposide:Control) for each cell line and data from 2 independent etoposide experiments were merged. Volcano plots were generated by plotting the -Log10(q-value) against Log2(FPKM ratio) using Microsoft Excel software. The data summarized in this publication can be found in Supporting Information described below.

## Immunoblotting and quantification

Detached cells were collected from 6-well plates and combined with adherent cells in PBS containing EDTA, centrifuged, washed with PBS, and centrifuged again to ensure collection of all live and dead material. Cell pellets were resuspended in lysis buffer (20mM HEPES pH 7.4, 100mM NaCl, 1% Triton X-100, 1mM $Na_3VO_4$, 1mM NaF, plus 1mM PMSF, and 10μg/ml each of aprotinin, leupeptin, pepstatin, and chymostatin), diluted in SDS-PAGE sample buffer, boiled, centrifuged, and subjected to SDS-PAGE before transfer to nitrocellulose membranes (GE Healthcare). Membranes were blocked in PBS + 5% milk (PBS-M) before being probed with primary antibodies (S3 Table) diluted in PBS-M overnight at 4°C plus an additional 2-3h at room temperature. Membranes were rinsed twice with PBS and washed thrice with PBS + 0.5% Tween-20 (PBS-T). Membranes were then probed with secondary antibodies

conjugated to IRDye-800, IRDye-680, or horseradish peroxidase (S3 Table) and diluted in PBS-M. Membranes were again rinsed with PBS and washed with PBS-T. Blots were imaged using a LI-COR Odyssey Fc imaging system. Band intensities were determined using the Analysis tool in LI-COR Image Studio software, and quantities of proteins-of-interest were normalized to tubulin, actin, and/or GAPDH loading controls.

### Apoptosis and caspase activation assays

For live fluorescence-based assays, approximately $2.5 \times 10^5$ cells were seeded into 24-well glass-bottom plates and allowed to grow for 24h prior to DMSO or etoposide treatments. For end-point apoptosis assays, Alexa488-AnnexinV (Invitrogen), Propidium Iodide (Invitrogen), and Hoechst (Thermo Scientific) were added directly to the media (S3 Table) and incubated for 15min at 37˚C in 5% $CO_2$. For caspase activation assays, CellEvent Caspase-3/7 Detection Reagent (Invitrogen) and Hoechst were added to the media and incubated for 30min. All live imaging was performed at 37˚C as described below. For mitochondrial visualization in Fig 9, MitoTracker Red (Invitrogen) was added for 30min prior to fixation.

### Immunostaining assays

For immunofluorescence microscopy, approximately $2.5 \times 10^5$ cells were seeded onto 12mm glass coverslips in 24-well plates and allowed to grow for 24h. After DMSO or etoposide treatments, cells were washed with PBS, fixed in 2.5% or 3.7% paraformaldehyde (PFA) in PBS for 30min, washed, permeabilized with 0.1% TritonX-100 in PBS, washed, and incubated in blocking buffer (1% FBS + 1% bovine serum albumin (BSA) + 0.02% $NaN_3$ in PBS) for a minimum of 15min. Cells were probed with primary antibodies (S3 Table) diluted in blocking buffer for 45min. Cells were washed and treated with AlexaFluor-conjugated secondary antibodies, DAPI, and/or AlexaFluor-conjugated phalloidin (S3 Table) for 45min, followed by washes and mounting in Prolong Gold anti-fade reagent (Invitrogen).

### Fluorescence microscopy

All live and fixed images were captured using a Nikon Eclipse T*i* inverted microscope with Plan Apo 100X/1.45, Plan Apo 60X/1.40, or Plan Fluor 20x/0.5 numerical aperture objectives using an Andor Clara-E camera and a computer running NIS Elements software. Live cell imaging was performed in a 37˚C chamber (Okolab). All cells were viewed in multiple focal planes, and Z-series were captured at 0.2–0.4μm steps. Images presented in the figures represent one slice or two-slice projections. All images were processed and/or analyzed using ImageJ [103].

### Image processing and quantification

The ImageJ Cell Counter plugin was used for live apoptosis or caspase-3/7 activation assays, and for fixed cytochrome *c*, cleaved caspase-3, or JMY staining assays by manually counting the total number of cells in the Hoechst or DAPI channel and the number of cells that were positive for AnnV fluorescence, AnnV fluorescence plus nuclear PI fluorescence, nuclear PI fluorescence, nucleic acid-associated DEVD reporter fluorescence, cytosolic non-mitochondrial cytochrome *c* staining, clustered cleaved caspase-3 staining, and cytosolic punctate cytochrome *c* or JMY staining. For analyses of nuclear γH2AX, p21, JMY, or actin levels, the Threshold, Watershed, and Analyze tools were used in the DAPI channel to separate individual nuclei, and the ROI Manager Measure tool was used in the γH2AX, p21, JMY, or actin channels to measure the mean fluorescence intensity per nucleus. To count the number of

γH2AX foci per nucleus, the Find Maxima and ROI Manager Measure tools were used in the γH2AX channel. For analyses of total cellular p53, JMY, or actin levels, the Selection tool was used in the phalloidin channel to select individual cells, and the Measure tool was used in the p53, JMY, or actin channel to measure the mean fluorescence intensity per cell. For analyses of cytoplasmic JMY or actin levels, the nuclear intensity was subtracted from the total intensity for individual cells. Pixel intensity plots were generated using the Plot Profile tool. Lines were drawn through mitochondria- or JMY-rich regions after identification in the AIF/MitoTracker or JMY channels.

## Reproducibility and statistics

All conclusions were based on observations made from at least 3 separate experiments and quantifications were based on data from 3–5 representative experiments, except where noted in the Figure Legends. To capture a breadth of apoptotic phenotypes, multiple fields-of-view containing cells at different densities were imaged. The sample size used for statistical tests was the number of times an experiment was performed, except where noted in the Legends. Statistical analyses were performed using GraphPad Prism software. Statistics on data sets with 3 or more conditions were performed using ANOVAs followed by Tukey's post-hoc test unless otherwise indicated. P-values for data sets comparing 2 conditions were determined using unpaired t-tests unless otherwise noted. P-values $<0.05$ were considered statistically significant. Numerical data underlying graphs or summary statistics are shown in S1 and S2 Datasets.

## Supporting information

**S1 Table. Cell Lines.**
(PDF)

**S2 Table. RNA and DNA.**
(PDF)

**S3 Table. Immunofluorescence and immunoblotting reagents.**
(PDF)

**S1 Fig. Analyses of WASP-family knockout cell lines uncover roles for JMY and WHAMM in apoptosis. (A)** HAP1, WAVE1[KO], WAVE2[KO], WAVE3[KO], WAVE Complex[KO] (BRK1[KO]), N-WASP[KO], Cortactin[KO], and WASH Complex[KO] (CCDC53[KO]) cells were collected, lysed, and immunoblotted with different combinations of antibodies to WAVE1, WAVE2, N-WASP, Cortactin, WASH, and Strumpellin. Tubulin, actin, and GAPDH were used as loading controls. The relative quantities of WASP-family proteins in each cell line (shown beneath their respective blots) were determined by densitometry and normalization to the loading controls in these representative experiments confirming the WASP-family member deficiencies. **(B)** HAP1, WAVE1[KO], WAVE2[KO], WAVE3[KO], and WAVE Complex[KO] cells were fixed and stained with phalloidin (F-actin; green) and DAPI (DNA; blue). Scale bar, 100μm. Note the rounder morphologies of the WAVE Complex[KO] cells compared to the other cells. **(C)** Parental (HAP1, eHAP) and WASP-family knockout (JMY[KO-1A], WHAMM[KO-2], N-WASP[KO], WAVE1[KO], WAVE2[KO], WAVE3[KO], WAVE Complex[KO], WASH Complex[KO], Cortactin[KO]) cells were treated with DMSO or 5μM etoposide for 6h and stained with Alexa488-AnnexinV (AnnV; green), Propidium Iodide (PI; magenta), and Hoechst (DNA; blue). Scale bar, 100μm. These panels depict some of the imaging data that was incorporated into the summary graph in Fig 1.
(TIF)

**S2 Fig. Cells specifically lacking the WASP-family members JMY or WHAMM undergo less apoptosis following DNA damage. (A)** Parental (HAP1, eHAP) and WASP-family knockout (JMY$^{KO-1A}$, WHAMM$^{KO-2}$, WAVE1$^{KO}$) cells were treated with DMSO or 5μM etoposide for 6h and stained with Alexa488-AnnV (green), PI (magenta), and Hoechst (DNA; blue). Scale bar, 100μm. These panels display the individual grayscale channels that comprise the merged images in [Fig 1]. **(B-C)** Representative examples of AnnV-negative/PI-negative (AnnV$^-$PI$^-$) non-apoptotic, AnnV-positive/PI-negative (AnnV$^+$PI$^-$) early apoptotic, AnnV/PI double-positive (AnnV$^+$PI$^+$) late apoptotic, or AnnV-negative/PI-positive (AnnV$^-$PI$^+$) necrotic cells are shown. Arrowheads in C highlight examples of hoechst-stained DNA condensation and nuclear fragmentation in HAP1 cells. Scale bars, 25μm.
(TIF)

**S3 Fig. JMY, WHAMM, RhoD, and WHAMM/JMY knockout cell lines contain loss-of-function mutations derived from frameshifts or altered splicing. (A)** HAP1 cells were CRISPR/Cas9 engineered using guide RNAs to the first or second exon of the *JMY* gene. A 17bp deletion in JMY$^{KO-1A}$, a 10bp deletion in JMY$^{KO-1B}$, and a 2bp deletion in JMY$^{KO-2}$ resulted in frameshifts and premature stop codons. **(B)** eHAP cells were treated with guide RNAs to the second or fourth exon of the *WHAMM* gene. A 10bp deletion in WHAMM$^{KO-2}$ and a 7bp deletion in WHAMM$^{KO-4}$ resulted in frameshifts and premature stop codons. **(C)** eHAP cells were treated with guide RNAs to the second exon of the *RHOD* gene. A 22bp deletion in RhoD$^{KO-2A}$ is predicted to result in defective splicing (not shown) and/or a frameshift (shown), while a 1bp deletion in RhoD$^{KO-2B}$ results in a simple frameshift and premature stop codon. **(D)** WHAMM$^{KO-2}$ cells were treated with guide RNAs to the first or second exon of the *JMY* gene. A 16bp deletion in WHAMM/JMY$^{DKO-1}$ and a 35bp deletion in WHAMM/JMY$^{DKO-2}$ resulted in frameshifts and premature stop codons.
(TIF)

**S4 Fig. Parental and knockout cell lines exhibit high levels of γH2AX expression and foci in response to DNA damage. (A)** HAP1, eHAP, and knockout cell lines were treated with DMSO or 5μM etoposide for 6h before immunoblotting with anti-γH2AX and anti-tubulin antibodies. **(B)** HAP1, eHAP, and JMY$^{KO}$ cells were treated with DMSO or etoposide, fixed, and stained with a γH2AX antibody (green), phalloidin (F-actin; magenta), and DAPI (DNA; blue). Representative images show increased nuclear γH2AX foci upon etoposide treatment. Scale bar, 25μm. **(C)** Nuclear γH2AX fluorescence intensity was calculated using ImageJ (n = 48–70 nuclei per sample from a representative experiment). **(D)** The number of γH2AX foci per nucleus was determined using ImageJ (n = 54–81 nuclei per sample from a representative experiment). Significance stars are in reference to the etoposide-treated eHAP cell line. Fewer γH2AX foci were observed in etoposide-treated RhoD$^{KO}$ samples because of the loss of some dead cells prior to fixation to the slide. $^*$p<0.05 (ANOVA, Tukey post-hoc tests).
(TIF)

**S5 Fig. Transient WHAMM or JMY depletion in multiple cell lines results in less apoptosis. (A)** eHAP or HAP1 cells were treated with control siRNAs or independent siRNAs for the WHAMM gene before collecting RNA and performing RT-PCR with primers for *WHAMM* and *GAPDH*. **(B)** The cells were treated with DMSO or 5μM etoposide for 6h and stained with Alexa488-AnnV, PI, and Hoechst. The % of AnnV-positive cells was calculated and each bar represents the mean ±SD from 3 experiments (n = 1,998–4,697 cells per bar). Significance stars refer to comparisons to the siControl samples. **(C)** U2OS or HeLa cells were treated with control siRNAs or independent siRNAs for the JMY gene before immunoblotting with antibodies to JMY and tubulin. **(D)** The cells were treated with DMSO or 10μM etoposide for 6h

and stained with Alexa488-AnnV (green), PI (magenta), and Hoechst (blue). Scale bar, 100μm. **(E)** The % of AnnV-positive cells was calculated as in panel (B) (n = 476–866 cells per bar). **(F)** JMY band intensities on immunoblots were normalized to tubulin bands and plotted against the % of AnnV-positive cells. Each point represents the mean ±SD from 3 images in a given experiment (n = 69–440 cells per point). The slopes in the linear trendline regression equations for etoposide-treated samples (U2OS: Y = 38.31X - 3.86; HeLa: Y = 35.16X + 7.13) were significantly non-zero (p<0.001, $R^2$>0.74). AU = Arbitrary Units. $^{**}$p<0.01; $^{***}$p<0.001 (ANOVA, Tukey post-hoc tests).
(TIF)

**S6 Fig. Transient depletion of p53 results in less apoptosis in eHAP and WHAMM$^{KO}$ cells following DNA damage. (A)** eHAP and WHAMM$^{KO-2}$ cells were treated with control siRNAs or independent siRNAs for the TP53 gene before immunoblotting with antibodies to p53 and tubulin. **(B)** The cells were treated with DMSO or 5μM etoposide for 6h and stained with Alexa488-AnnV, PI, and Hoechst. The % of AnnV-positive cells was calculated and each bar represents the mean ±SD from 3 experiments (n = 2,212–3,846 cells per bar). **(C)** p53 band intensities on immunoblots were normalized to tubulin bands and plotted against the % of AnnV-positive cells. Each point represents the mean ±SD from 3 fields-of-view in a given experiment (n = 576–1,509 cells per point). The slopes in the linear trendline regression equations for etoposide-treated samples (eHAP: Y = 35.44X + 10.99; WHAMM$^{KO}$: Y = 13.28X + 11.81) were significantly non-zero (p<0.001, eHAP: $R^2$ = 0.87; WHAMM$^{KO}$: $R^2$ = 0.47). AU = Arbitrary Units. These studies accompanied the experiments in Fig 5. $^{**}$p<0.01; $^{***}$p<0.001 (ANOVA, Tukey post-hoc tests).
(TIF)

**S7 Fig. JMY is not required for p53 stabilization, nuclear accumulation, phosphorylation, or acetylation. (A)** HAP1 and JMY$^{KO}$ cells were treated with DMSO or 5μM etoposide for 6h before immunoblotting with antibodies to p53, p53$^{phospho-S15}$, p53$^{acetyl-K382}$, p53$^{phospho-S46}$, GAPDH, and tubulin. **(B)** HAP1 and JMY$^{KO}$ cells were treated with DMSO or etoposide for 6h before being fixed and stained with a p53 antibody (green) and DAPI (DNA; blue). Scale bar, 30μm. **(C)** Cellular p53 fluorescence intensities were measured in ImageJ and the total p53 intensity was normalized to the HAP1 DMSO sample. Each bar represents the mean ±SD from 3 experiments (n = 196–343 cells per timepoint). AU = Arbitrary Units.
(TIF)

**S8 Fig. JMY-knockout cells undergo a prolonged p21-associated proliferation arrest instead of rapid apoptosis following DNA damage. (A-C)** HAP1 and JMY$^{KO}$ cells were treated with DMSO or 5μM etoposide for 6h before washout (triangles) and replacement with regular media. Samples were stained with Alexa488-AnnV, PI, and Hoechst at the indicated time points. In (A-B), the total # of cells (live and dead) was counted in ImageJ and normalized to the # at 0h for each sample. The total # of cells is displayed as the fraction of live AnnV-negative (AnnV$^-$) or apoptotic AnnV-positive (AnnV$^+$) cells (n = 1,461–9,108 cells per bar for DMSO; n = 1,415–2,992 cells per bar for etoposide). Panels (A) and (B) depict the same data in two different formats. In (C), the % of AnnV-positive cells was calculated and each point represents the mean ±SD from 3 experiments. Significance stars refer to comparisons among the etoposide-treated samples. RU = Relative Units. **(D)** HAP1 and JMY$^{KO}$ cells were treated with DMSO or etoposide for 6h before collecting RNA and performing RT-PCR with primers for *CDKN1A* and *GAPDH*. Agarose gel band intensities were quantified in ImageJ, and values for *CDKN1A* were normalized to *GAPDH* and plotted in the adjacent bar graph. AU = Arbitrary Units. **(E)** Cells were treated with DMSO or etoposide for 6h before being fixed and stained

with a p21 antibody (green) and DAPI (DNA; blue). Scale bar, 30μm. **(F)** Nuclear p21 fluorescence intensity was measured in ImageJ, and each point represents the mean ±SD from 3 fields-of-view in a representative experiment (n = 216–418 cells per point). $^{**}$p<0.01; $^{***}$p<0.001 (ANOVA, Tukey post-hoc tests).
(TIF)

**S9 Fig. Expression of RhoD is turned on in JMY-knockout cells, and etoposide treatment does not cause a differential upregulation of pro-apoptotic genes. (A)** RNA collected from HAP1 and JMY$^{KO}$ cells was subjected to mRNA sequencing analysis. FPKM values for individual genes with expression differences in JMY$^{KO-1A}$ vs HAP1 samples of at least 2-fold and with a significance q-value of <0.05 are shown in dark red (turned on), pink (up-regulated), light blue (down-regulated), or dark blue (turned off). Each bar represents the mean value ±SD from 3 independent RNA samples per genotype. **(B)** RNA collected from HAP1 and JMY$^{KO}$ cells treated with 5μM etoposide for 6h was subjected to mRNA sequencing analysis and comparisons to control samples from (A). FPKM values for individual genes with expression differences of at least 2-fold and with a significance q-value of <0.05 in etoposide vs control-treated HAP1 (gray bars) or JMY$^{KO-1A}$ (purple bars) samples are shown. Each bar represents the mean value ±SD from 2 independent RNA samples per genotype. Bolded black gene names indicate a >2-fold expression difference in both HAP1 and JMY$^{KO}$ cells, purple gene names indicate a >2-fold expression difference in JMY$^{KO}$ cells only, and gray gene names indicate a >2-fold expression difference in HAP1 cells only. Stars indicate differences of >2 in Log2(FPKM) values between HAP1 and JMY$^{KO}$ cells, with *IL8* (in one replicate) and *CCL20* (in both replicates) upregulated more in HAP1 cells, and *TNF*, *EGR1*, *JUN*, *PLK2*, and *DDIT4* upregulated more in JMY$^{KO}$ samples.
(TIF)

**S10 Fig. Expression of genes encoding actin nucleation factors, cell cycle arrest proteins, canonical apoptosis regulators, and representative small G-proteins.** RNA collected from HAP1 and JMY$^{KO}$ cells was subjected to mRNA sequencing analysis as in Fig 6. Data for several classes of gene products involved in apoptosis or cytoskeletal regulation are shown. For etoposide-treated samples, genes related to TNF or other signaling pathways are also included.
(PDF)

**S11 Fig. RhoD-deficient cells display more frequent cytochrome *c* release, exhibit more caspase cleavage, and undergo more apoptosis following DNA damage. (A)** JMY$^{KO-1A}$ cells were treated with control siRNAs or independent siRNAs for the RHOD gene before performing RT-PCRs with primers to *RHOD*, *GAPDH*, and *β-ACTIN*. **(B)** Cells were treated with DMSO or 5μM etoposide for 6h and stained with Alexa488-AnnV, PI, and Hoechst. The % of AnnV-positive cells was calculated and displayed as the fraction of AnnV-positive/PI-negative (AnnV$^+$PI$^-$) or AnnV/PI double-positive (AnnV$^+$PI$^+$) cells. Significance stars refer to comparisons of total AnnV$^+$ counts for siControl vs siRhoD samples. The % of AnnV-negative/PI-positive (AnnV$^-$PI$^+$) necrotic cells was also quantified. Each bar represents the mean ±SD from 3 experiments (n = 3,166–4,998 cells per sample). **(C)** *RHOD* band intensities were normalized to *β-ACTIN* and plotted versus the % of AnnV-positive cells. Each point represents the mean ±SD from 3 fields-of-view in a given experiment (n = 670–1,919 cells per point). **(D)** eHAP and RhoD$^{KO}$ cells were treated with DMSO or etoposide and stained with Alexa488-AnnV (green), PI (magenta), and Hoechst (DNA; blue). Scale bar, 100μm. **(E)** Cells were treated with DMSO or etoposide, fixed, and stained with antibodies to detect cytochrome *c* (Cyto *c*; green), AIF (Mito; magenta), and DAPI (DNA; blue). Scale bar, 25μm. **(F-G)** Cells were treated with DMSO for 6h or etoposide for 3 or 6h, and extracts were immunoblotted with antibodies to

caspase-9 (Casp-9$^{Pro}$ and Casp-9$^{Cleaved}$), caspase-3 (Casp-3$^{Cleaved}$), and tubulin. **(H)** eHAP and RhoD$^{KO}$ cells were treated with DMSO or etoposide for 6h and stained with caspase-3/7 green detection reagent (Cleaved DEVD; green) and Hoechst (DNA; blue), or fixed and stained with an antibody that recognizes active caspase-3 cleaved at Asp175 (Cleaved Casp-3; green) and DAPI (DNA; blue). Scale bars, 100μm. $^*$p$<$0.05, $^{**}$p$<$0.01 (ANOVA, Tukey post-hoc tests). (TIF)

**S12 Fig. F-actin is reorganized in the cytosol of HAP1 cells in response to DNA damage.** **(A)** HAP1 cells were treated with 5μM etoposide for 6h before being fixed and stained with a JMY antibody (green), a cyto *c* antibody (magenta), and DAPI (DNA; blue). Arrowheads and magnifications represent examples of juxtanuclear JMY and cyto *c* puncta. Scale bars, 25μm, 10μm. **(B)** U2OS cells were treated with 10μM etoposide for 6h before being incubated with MitoTracker (Mito; cyan), fixed, and stained with a JMY antibody (green), a cyto *c* antibody (magenta), and phalloidin (F-actin; yellow). Arrowheads and magnifications depict clusters of JMY and cyto *c* puncta in an F-actin territory. Scale bars, 25μm, 10μm. **(C)** HAP1 and JMY$^{KO}$ cells were treated with 5μM etoposide for 6h before being fixed and stained with a JMY antibody, DAPI, and either phalloidin to visualize F-actin or an actin antibody to visualize total actin. Nuclear and cytoplasmic F-actin, total actin, and JMY fluorescence intensities were measured in ImageJ and the total intensity was normalized to the HAP1 DMSO sample. Each bar represents the mean ±SD from 2–4 experiments (F-actin: n = 106–163 cells per bar; Total actin: n = 103–148 cells per bar; JMY: n = 234–311 cells per bar). Gray significance stars refer to comparisons of the cytoplasmic intensity to the DMSO samples and black significance stars refer to comparisons of the nuclear intensity to the DMSO samples. AU = Arbitrary Units. $^*$p$<$0.05, $^{**}$p$<$0.01 (t-test). **(D)** Model for JMY, WHAMM, the Arp2/3 complex, F-actin, and RhoD in apoptosis. Line thickness reflects different degrees of apoptosis-related activities, line dashing represents a potential role for RhoD based on interactions shown in the literature. (TIF)

**S1 Dataset. File containing the data underlying the summary graphs.** (XLSX)

**S2 Dataset. File containing the data underlying the RNA-seq figures.** (XLSX)

## Acknowledgments

We thank Bo Reese at the UConn Center for Genome Innovation for guidance with RNA-seq and analysis, Vanessa Vlaun for help with microscopy quantification, Frida Zink for help processing knockout cells, and L.T. Bear for support with experimental design. We also thank Katrina Velle, Aoife Heaslip, and Campellone Lab members for their comments on this paper.

## Author Contributions

**Conceptualization:** Virginia L. King, Nathan K. Leclair, Kenneth G. Campellone.

**Data curation:** Virginia L. King, Nathan K. Leclair, Kenneth G. Campellone.

**Formal analysis:** Virginia L. King, Nathan K. Leclair, Kenneth G. Campellone.

**Funding acquisition:** Kenneth G. Campellone.

**Investigation:** Virginia L. King, Nathan K. Leclair, Alyssa M. Coulter, Kenneth G. Campellone.

**Methodology:** Virginia L. King, Nathan K. Leclair, Alyssa M. Coulter, Kenneth G. Campellone.

**Project administration:** Kenneth G. Campellone.

**Resources:** Kenneth G. Campellone.

**Supervision:** Kenneth G. Campellone.

**Validation:** Virginia L. King, Nathan K. Leclair, Alyssa M. Coulter, Kenneth G. Campellone.

**Visualization:** Virginia L. King, Nathan K. Leclair, Kenneth G. Campellone.

**Writing – original draft:** Virginia L. King, Kenneth G. Campellone.

**Writing – review & editing:** Virginia L. King, Nathan K. Leclair, Alyssa M. Coulter, Kenneth G. Campellone.

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
