## [Decision Letter · Decision Letter 0]

26 Aug 2020

Dear Ken,

Thank you very much for submitting your Research Article entitled 'The actin nucleation factors JMY and WHAMM enable a rapid p53-dependent pathway of apoptosis' to PLOS Genetics. Our apologies for the delay in reaching a decision, which was due to difficulty in securing appropriate reviewers.

The manuscript has now been fully evaluated by three independent peer reviewers. As you will see, the reviewers appreciated the attention to an important problem, but raised some substantial concerns about the current manuscript, including some of the experimental design and some of the data. Based on the reviews, we will not be able to accept this version of the manuscript, but we would be willing to review again a much-revised version. We cannot, of course, promise publication at that time.

If you decide to revise the manuscript for further consideration at PLOS Genetics, please aim to resubmit within the next 60 days, unless it will take extra time to address the concerns of the reviewers, in which case we would appreciate an expected resubmission date by email to plosgenetics@plos.org.

You can use this link to log into the system when you are ready to submit a revised version, having first consulted our Submission Checklist.

We are sorry that we cannot be more positive about your manuscript at this stage. Please do not hesitate to contact us if you have any concerns or questions.

Yours sincerely,

Irene Chiolo

Guest Editor

PLOS Genetics

Gregory Barsh

Editor-in-Chief

PLOS Genetics

Reviewer's Responses to Questions

**Comments to the Authors:**

Reviewer #1: The actin-nucleation factors JMY and WHAMM enable a rapid p53-dependent pathway of apoptosis

The purpose of this manuscript is to examine potential roles for nucleation promoting factors (NPFs) in programmed cell death. Utilizing Hap1 cells that were genetically depleted of different WASP-family proteins, the authors found that WHAMM and JMY are critical for apoptosis, regulating early steps of cytochrome c release from mitochondria and the caspase-cleavage cascade. The small GTPase RhoD was found to be upregulated in JMY KO cells, and loss of RhoD in Hap1 cells enhanced apoptotic responses to DNA damage, suggesting that RhoD normally promotes cell survival and counterbalances the pro-apoptotic function of JMY.

Overall, this study expands our understanding of apoptosis by identifying two new players in programmed cell death, suggesting a direct link between the actin cytoskeleton and the intrinsic apoptotic pathway. The experiments are thoughtful and straightforward, and direct conclusions are supported by the data. However, the main function of NPFs is the activation of the actin nucleator Arp2/3-complex, the relevance of which has not been addressed in this study. The questions, whether the regulation of apoptosis by WHAMM and JMY is actin-dependent or -independent, and whether both NPFs act in concert in the same pathway or function separately, have been left unanswered. Thus, while I think the manuscript is worthy of publication, a few issues, listed below in more detail, should be addressed beforehand.

Major comments:

1. This study has not addressed whether JMY and WHAMM have specific or redundant functions in apoptosis. Thus, it would be useful to subject JMY/WHAMM double KO cells to Etoposide-treatment and determine the rate of apoptosis in such a cell line. I understand that generating a double KO cell line is time-consuming. Therefore, the authors could alternatively employ one of their JMY siRNAs in WHAMM KO cells.

2. Another open question is whether the apoptotic function of JMY and WHAMM is Arp2/3-complex/actin-dependent. I suggest performing RNA-mediated silencing of Arp2 or Arp3 (leading to the loss of the entire complex) and analyzing the rate of apoptosis in these cells. The result might reveal a role for Arp2/3-complex in DNA damage-induced apoptosis, indicating an Arp2/3-dependent role for JMY and WHAMM in this process.

3. I am aware that rescue experiments are not trivial in general, however, it would be interesting to re-express JMY wt versus a JMY-WCA mutant in JMY KO cells and test for efficiency of apoptosis. Along with point 2., this could shed light on the relevance of JMY’s function as an Arp2/3-complex activator for apoptosis. Furthermore, it would be a good control experiment to show that JMY wt is able to rescue the observed apoptosis phenotype.

4. I assume that the authors had problems to find a functional/specific RhoD antibody, which is why they used RT-PCR to examine the transcript levels of this protein in cells. Nonetheless, it would be of major advantage for the manuscript to show the increase of endogenous RhoD in JMY KO cells at the protein expression level. Given that mRNA levels of RhoD are undetectable in control Hap1 cells by RT-PCR, the determination of RhoD protein levels is even more relevant. Furthermore, the authors have not analyzed RhoD expression in WHAMM KO cells, although it would be interesting to analyze whether the increased expression of RhoD is a general cell response in order to cope with apoptosis defects, especially in regard to WHAMM being an interactor of RhoD.

Minor comments:

Fig 1A: I would prefer that the authors show representative images for all KO cell lines, even the ones that had no effect on apoptosis.

Fig 4E and F: The marker band for ‘Casp-3 cleaved’ is depicted as 20 kDa in E and 25 kDa in F.

Fig 6B: This is a rather modest effect for siTP53, which indicates that Etoposide-induced apoptosis is not entirely driven by p53. Is there a p53-independent apoptotic pathway? And can the authors exclude that JMY is also functioning in p53-independent apoptotic pathways? Comment on that.

Fig 6F and G: Why were only control Hap1 cells examined for nuclear translocation of p53? Although the figure title says that ‘JMY is not required for nuclear accumulation of p53’, JMY KO cells are not shown in part F and G.

Fig 7A: What is (RU)? I assume the authors mean AU?

S1Table: The BRK1 KO cell line is referred to as WAVE complex KO, which I have a hard time to believe from what is known about the WAVE complex. Deletion of components, such as Sra1, Nap1 or WAVE have been shown to significantly affect the expression of the entire complex. However, the small component BRK1 (~10 kDa) most likely does not affect the expression of the remaining components. Do the authors have data to confirm that the expression of the entire complex is affected by loss of BRK1?

Line 525: remove ‘in’

Reviewer #2: The manuscript here reports a role of the actin nucleation factors JMY and WHAMM in etoposide-induced apoptosis. Using knockdown and knockout approach, it was shown that deletion/depletion of JMY and WHAMM, as well as depletion of p53, reduced the etoposide-induced apoptosis. RNA-seq analysis revealed increased levels of RhoD in JMY knockout cells. Furthermore, knockout of RhoD sensitized cells to etoposide treatment, suggesting it may be the pro-survival factor in the JMY-deleted cells. In all, despite that the results in general supported a role of JMY and WHAMM in etoposide-induced apoptosis, the physical and functional links with either p53 or RhoD were not established. More convincing evidence is required.

1. Fig. 2 and 3 are redundant. Experiments with siRNA knockdown (Fig. 3) can be moved to supplementary data.

2. There are many post-translational modification sites in p53 that may have differential impacts on p53-targeted promoters. For apoptosis, pS46 may be more relevant than pS15. The involvement of pS46 should be investigated (Fig. 6). Furthermore, induction of apoptosis-related genes, such as Bax and PUMA, should be compared.

3. The authors suggested that cell arrest may have occurred in etoposide-treated JMY KO cells (Fig. 7). Flow cytometry analysis is needed to verify this point. Induction of CDKN1A can hardly address this point since HAP1 cells undergo apoptosis despite similar amount of CDKN1A induction.

4. RNA-seq appeared to be conducted with samples collected from untreated cells. As DNA damage response is being investigated, it may be more appropriate to compare untreated and etoposide-treated HAP1 and JMY KO cells.

5. RT-PCR products shown in Fig. 8C and 8E were mostly overly amplified (except RhoD and WHAMM), which could easily mask the difference. Reduced amplification cycle should be tested, or alternatively, qPCR can be used.

6. Although RhoD expression is elevated in JMY KO cells, data presented did not support its involvement in the reduced apoptosis seen in JMY o WHAMM KO cells. Depletion of RhoD did not exacerbate etoposide-induced apoptosis in JMY KO cells, suggesting they may not function in the same axis (Fig. 9C). As for its role in WHAMM related apoptotic pathway, it may be more appropriate to use the WHAMM KO eHAP cells (Fig. 9E-9I).

7. Fig. 10 nicely showed that RhoD is a pro-survival factor under etoposide treatment in eHAP1 cells. However, it also has not addressed the relationship with JMY or WHAMM. It can be presented as a supplementary data for Fig. 9.

Minor points:

1. At some points, JMY KO cells were referred as JMY mutant cells, which could be easily confused with point mutant.

2. Discussion is a bit too long.

Reviewer #3: King and colleagues report that two proteins known to function in actin assembly have roles in promoting apoptosis. Modulators of actin dynamics and actin itself are now known to medicate apoptosis, leading to the view that actin cytoskeleton is a biosensor that integrates internal and external apoptotic stimuli. But these studies typically focus on proteins that depolymerize or cross-link actin. The current study is on proteins that promote actin polymerization, thus filling a knowledge gap. While the study is suitable for PLoS Genetics in principle, problems with the experimental design and some of the data prevent me from recommending publication. The data that JMY- or WHAMM-depleted cells show reduced caspase activation and reduced apoptosis after etoposide treatment (Figs 1-4) are convincing, as are the (negative) data on the role of JMY in p53 activation (Fig. 6). Problematic data/conclusions are:

1. Conclusions made based on data from different cell lines. For example, the effect JMY depletion was studied in Hap1 (nearly haploid) but the effect of WHAMM depletion was studied in eHap (fully haploid, derived from Hap1 by CRISPR). These cell lines behave differently as seen later in RhoD experiments, yet the authors make conclusions across cell lines. For example, JMY was concluded to be more important for apoptosis than WHAMM (line 150 and models in Fig. 10). To make such a conclusion, knocked down phenotypes must be compared in the same cell line.

2. If proteins being tested are in a complex, why did depletion of just one or two members showed a phenotype? The authors should confirm the success of the knockout for genes that did not give a phenotype, by Western blotting like they did for genes that did give a phenotype.

3. Cytochrome C data in Fig. 5 is not convincing. I cannot see the difference between Hap1 and JMY ko (drug-treated) in A nor between eHap and WHAMM ko (drug treated cells) in B, yet quantifications in C and D show large differences. How exactly are the images quantified? It seems to be a subjective YES/NO call for whether cytochrome C is mitochondrial or cytoplasmic but how is that call being made? Along the same lines, Figure S3D reports the number of gamma-H2AX foci, but based on the images in B, I do not see how the authors can quantify foci number, especially in drug-treated cells. It is very hard to see discrete foci. The Western blot data in S3A is convincing, however.

4. Figure 7 is perplexing. The authors set up this figure by saying that inhibition of cell proliferation is an early response to DNA damage (line 234), yet when they address this experimentally, they are not looking at the early pause in the cell cycle (presumably checkpoint-mediated) but what happens over ~2 days after the drug has been washed out. Here, they see an ~50% increase in Hap1 cell number at the 6h time point compared to the 0h time point, which does not make sense because at 5uM of drug used in this experiment, 40% of Hap1 cells should be dead (Fig. 2C) and the rest arrested in the cell cycle by checkpoints. Over the next 48hr, the relative number of Hap1 cells stayed the same but % that is Annexin 5+ increased to reach 100%. Why does the cell number stay the same when larger and larger fractions of them are dying? Are they reaching a steady state because proliferation is off-set by death? Are dead cells included in the cell count? Are cells showing signs of apoptosis but not really dead? It is important to understand what is happening in Hap1 controls in this experiment so that the data for JMY knock down cells can be interpreted. If conclusions are to be made about ‘cell cycle arrest’, cell cycle status should be examined directly.

5. While RhoD RNA is induced in JMY ko Hap1 cells, depletion of it makes little difference to apoptosis. So, the reason for reduced apoptosis in JMY ko cells is still unknown. Depletion of RhoD in eHap cells did increase apoptosis. Why don’t the authors study the effect of JMY depletion on apoptosis and RhoD expression in eHap cells? They went as far as confirming the role of JMY in Hela and U2OS cells but not in eHap. If the results in eHap cells are positive, then they can ask if increased RhoD is the reason JMY ko cells show increased apoptosis. Again, because the requirement for different genes are shown in different cell lines, the conclusions in Fig. 10F are not justified by the data. In fact, this major problem with the experimental design prevents the authors from making impactful conclusions.

**Have all data underlying the figures and results presented in the manuscript been provided?**

Reviewer #1: Yes

Reviewer #2: Yes

Reviewer #3: Yes

PLOS authors have the option to publish the peer review history of their article (what does this mean?). If published, this will include your full peer review and any attached files.

Reviewer #1: No

Reviewer #2: No

Reviewer #3: No

---

## [Decision Letter · Decision Letter 1]

1 Mar 2021

Dear Dr. Campellone,

Thank you very much for submitting your Research Article entitled 'The actin nucleation factors JMY and WHAMM enable a rapid p53-dependent and Arp2/3 complex-mediated pathway of apoptosis' to PLOS Genetics.

The manuscript was fully evaluated at the editorial level and by independent peer reviewers. The reviewers appreciated the attention to an important topic but identified some additional concerns that we ask you address in a revised manuscript.

We therefore ask you to modify the manuscript according to the review recommendations. Your revisions should address the specific points made by each reviewer. We believe that the remaining concerns can be addressed to our satisfaction by text changes, and will not require the addition of new data. We recommend a particular attention to the following points:

- With the added data, it is becoming more clear that the pathway controlled by JMY and WHAMM is independent of and is different from that by p53, yet the title still indicates that p53 and JMY/WHAMM work together in etoposide-induced apoptosis.

- The RhoD DN data in the new Fig. 10 is also conflicting as it decreased the JMY puncta. 

- The authors now provide new data in Fig. 8F, 9 showing that JMY sometimes appeared in bright, juxtanuclear clusters that seemed to overlap with cleaved caspase-3, cyto c and F-actin staining. This association of JMY-positive structures with cyto c and F-actin provides an interesting new data set for the overall conclusions in the manuscript. However, while the association of JMY with a subset of cyto c puncta is somewhat visible in Fig. 9C and 9G, there is clearly no co-localization of JMY and cyto c in Fig. 9E, on the contrary the signals are mutually exclusive. Similarly, in Fig. S12 cyto c and JMY are not overlapping in those clusters, also actin is not directly associating with either one of them. Therefore, I would strongly advice to remove Fig. 9E (and S12A) and overall soften statements about overlap/co-localization of these factors.

[LINK]

Yours sincerely,

Irene Chiolo

Guest Editor

PLOS Genetics

Gregory Barsh

Editor-in-Chief

PLOS Genetics

Reviewer's Responses to Questions

**Comments to the Authors:**

Reviewer #1: The authors showed that both loss of JMY and WHAMM decreased p53-dependent apoptosis in cells but did not address the relationship of both NPFs in this pathway. Here, in the revised version they included evidence for JMY and WHAMM acting non-redundantly in the same apoptotic pathway. Interestingly, the additional KO of JMY in WHAMM KO cells caused a decrease in Etoposide-induced apoptosis to the same extent as JMY KO alone, indicating that JMY is more potent and their effects are non-additive. Furthermore, the authors compared permanent loss of JMY/WHAMM with transient siRNA-mediated depletion of both factors with similar results. This is an important control experiment and I agree with the authors response to reviewer #2, that permanent, long-term and instant, transient protein removal might have diverse effects and needed to be tested.

New Fig. 8 adds an important point to the manuscript, namely that JMY/WHAMMs roles in apoptosis are Arp2/3- and thus actin-dependent. This conclusion might seem trivial as these proteins are first and foremost nucleation-promoting factors. However, since JMY also has functions in transcriptional regulation, it was important to clarify if its particular role in apoptosis is Arp2/3 complex-dependent or -independent.

I also appreciate the addition of new Fig. 10, which shows that expressed RhoD can be found around JMY/cyto c containing clusters, providing more evidence for an implication of RhoD in this pathway, besides the elevated expression found in 2 of the 3 JMY KO clones.

Taken together, the authors have sufficiently addressed all my comments and strengthened the manuscript with new findings. Thus, while I generally support publication, I would like to point out a few things regarding the new data.

The authors now provide new data in Fig. 8F, 9 showing that JMY sometimes appeared in bright, juxtanuclear clusters that seemed to overlap with cleaved caspase-3, cyto c and F-actin staining. This association of JMY-positive structures with cyto c and F-actin provides an interesting new data set for the overall conclusions in the manuscript. However, while the association of JMY with a subset of cyto c puncta is somewhat visible in Fig. 9C and 9G, there is clearly no co-localization of JMY and cyto c in Fig. 9E, on the contrary the signals are mutually exclusive. Similarly, in Fig. S12 cyto c and JMY are not overlapping in those clusters, also actin is not directly associating with either one of them. Therefore, I would strongly advice to remove Fig. 9E (and S12A) and overall soften statements about overlap/co-localization of these factors.

Reviewer #2: In this revised manuscript, additional data have been added and some concerns were addressed. The authors also added the Arp2/3 part to provide relevant mechanistic explanation. These old and new data, adding together, support the involvement of p53, JMY, WHAMM, RHOD, and Arp2/3 in regulating etoposide-induced apoptosis but unfortunately do not offer a coherent picture regarding the functional or physical connection between them. There are also issues related to interpretation and experimental designs. The manuscript should be restructured and may be reorganized to present these significant amounts of data. Examples of relevant issues are:

1. Although the manuscript started with examination of JMY and WHAMM, the majority body of data, experimental designs, and interpretations seemed to focus on JMY. This is further complicated by some of the results that suggest the effect of JMY and WHAMM may regulate apoptosis differently.

2. With the added data, it is becoming more clear that the pathway controlled by JMY and WHAMM is independent of and is different from that by p53, yet the title still indicates that p53 and JMY/WHAMM work together in etoposide-induced apoptosis.

3. The upregulation of RhoD in JMY KO cells may not convincingly explain the increased survival as it was not among the genes particularly enhanced by the etoposide treatment. The RhoD DN data in the new Fig. 10 is also conflicting as it decreased the JMY puncta. Besides, why wasn’t this exp. conducted in RhoD KO cells?

4. Additional evidence on how alteration of actin filament, either its polymerization or organization, affects release of cytochrome c or apoptosis could strengthen the Arp2/3 data.

Reviewer #3: The revised manuscript by King and colleagues addresses my concerns from the first round of review. New data on the role of Arp2/3 in apoptosis strengthens the story. I support the publication of this work.

**Have all data underlying the figures and results presented in the manuscript been provided?**

Reviewer #1: Yes

Reviewer #2: Yes

Reviewer #3: Yes

PLOS authors have the option to publish the peer review history of their article (what does this mean?). If published, this will include your full peer review and any attached files.

Reviewer #1: No

Reviewer #2: No

Reviewer #3: No

---

## [Editor Report · Decision Letter 2]

28 Mar 2021

Dear Dr. Campellone,

Sincere apologies for the delayed response. We are pleased to inform you that your manuscript entitled "The actin nucleation factors JMY and WHAMM enable a rapid Arp2/3 complex-mediated intrinsic pathway of apoptosis" has been editorially accepted for publication in PLOS Genetics. Congratulations! 

Yours sincerely,

Irene Chiolo

Guest Editor

PLOS Genetics

Gregory Barsh

Editor-in-Chief

PLOS Genetics

Comments from the reviewers (if applicable):

**Data Deposition**

http://datadryad.org/submit?journalID=pgenetics&manu=PGENETICS-D-20-00988R2

**Press Queries**

---

## [Editor Report · Acceptance letter]

8 Apr 2021

PGENETICS-D-20-00988R2 

The actin nucleation factors JMY and WHAMM enable a rapid Arp2/3 complex-mediated intrinsic pathway of apoptosis 

Dear Dr Campellone, 

We are pleased to inform you that your manuscript entitled "The actin nucleation factors JMY and WHAMM enable a rapid Arp2/3 complex-mediated intrinsic pathway of apoptosis" has been formally accepted for publication in PLOS Genetics! Your manuscript is now with our production department and you will be notified of the publication date in due course.

With kind regards,

Katalin Szabo

PLOS Genetics

On behalf of:
